# Creating Multi-Level Skill Hierarchies in Reinforcement Learning

**Joshua B. Evans**
Department of Computer Science
University of Bath
Bath, United Kingdom
jbe25@bath.ac.uk

**Özgür Şimşek**
Department of Computer Science
University of Bath
Bath, United Kingdom
o.simsek@bath.ac.uk

## Abstract

What is a useful skill hierarchy for an autonomous agent? We propose an answer based on a graphical representation of how the interaction between an agent and its environment may unfold. Our approach uses modularity maximisation as a central organising principle to expose the structure of the interaction graph at multiple levels of abstraction. The result is a collection of skills that operate at varying time scales, organised into a hierarchy, where skills that operate over longer time scales are composed of skills that operate over shorter time scales. The entire skill hierarchy is generated automatically, with no human intervention, including the skills themselves (their behaviour, when they can be called, and when they terminate) as well as the hierarchical dependency structure between them. In a wide range of environments, this approach generates skill hierarchies that are intuitively appealing and that considerably improve the learning performance of the agent.

## 1 Introduction

One of the most important open problems in artificial intelligence is how to make it possible for autonomous agents to develop useful action hierarchies on their own, without any input from humans, including system designers and domain specialists. Before addressing this algorithmic question, it is useful to first consider a conceptual one: *What constitutes a useful action hierarchy?* Here we focus on this conceptual question with the aim of providing a useful foundation for algorithmic development.

Our primary contribution is a characterisation of a useful action hierarchy. In defining this action hierarchy, we use no information other than a graphical representation of how the interaction between the agent and its environment may unfold. When this information is not known a priori, it would be discovered naturally by the agent as it operates in its environment. Beyond this interaction graph, no particular domain knowledge is needed. Hence, our approach is applicable broadly.

Multiple strands of earlier work have used the interaction graph as a basis for defining collections of actions. The main novelty in our approach is our use of *modularity maximisation* as a central organising principle to expose the structure of the interaction graph at multiple levels of abstraction. The outcome is an action hierarchy that enables the agent to explore its environment efficiently at multiple time scales.

Our approach yields an action hierarchy with four desirable properties. First, it contains actions that operate at a wide range of time scales. This is necessary to solve complex problems, which require agents to be able to act, learn, and plan at varying time scales. Secondly, the actions are naturally organised into a hierarchy, with actions that operate over longer time scales being composed of actions that operate over shorter time scales. This hierarchical structure offers substantial benefits

37th Conference on Neural Information Processing Systems (NeurIPS 2023).

over unstructured collections of actions. For example, it allows an agent to learn about not only the action it is currently executing but also all lower-level actions that are called in the process. In addition, a hierarchical structure allows actions to be updated in a modular fashion. For instance, any improvements to an action would be immediately reflected in all higher-level actions that call it. Thirdly, the action hierarchy is fully specified. This includes when each action can be selected for execution, how exactly it behaves, and when it terminates. It also includes the number of levels in the hierarchy and the exact dependency structure between the actions. Fourthly, the action hierarchy is generated automatically, with no human intervention.

In a diverse set of environments, the proposed approach translates into action hierarchies that are intuitively appealing. When evaluated within the context of reinforcement learning, they substantially improve learning performance compared to alternative approaches, with the largest performance improvement observed in the largest environment tested.

An important question for future research is how such an action hierarchy may be learned when the agent has no prior knowledge of the dynamics of the environment. We present an initial exploration of how this may be achieved, with positive results.

## 2 Background

We use the reinforcement learning framework, modelling an agent's interaction with its environment as a finite Markov Decision Process (MDP). An MDP is a six-tuple $(\mathcal{S}, \mathcal{A}, \mathcal{P}, \mathcal{R}, \mathcal{D}, \gamma)$, where $\mathcal{S}$ is a finite set of states, $\mathcal{A}$ is a finite set of actions, $\mathcal{P} : \mathcal{S} \times \mathcal{A} \times \mathcal{S} \to [0, 1]$ is a transition function, $\mathcal{R} : \mathcal{S} \times \mathcal{A} \times \mathcal{S} \to \mathbb{R}$ is a reward function, $\mathcal{D} : \mathcal{S} \to [0, 1]$ is an initial state distribution, and $\gamma \in [0, 1]$ is a discount factor. Let $\mathcal{A}(s)$ denote the set of actions available in state $s \in \mathcal{S}$. At decision stage $t$, $t \geq 0$, the agent observes state $s_t \in \mathcal{S}$ and executes action $a_t \in \mathcal{A}(s_t)$. Consequently, at decision stage $t + 1$, the agent receives reward $r_{t+1} \in \mathbb{R}$ and observes new state $s_{t+1} \in \mathcal{S}$. The *return* at decision stage $t$, denoted by $G_t$, is the discounted sum of future rewards, $G_t = \sum_{k=0}^{\infty} \gamma^k r_{t+k+1}$. A policy $\pi : \mathcal{S} \times \mathcal{A} \to [0, 1]$ specifies the probability of selecting action $a \in \mathcal{A}$ in state $s \in \mathcal{S}$. The objective is to learn a policy that maximises the expected return.

The *state transition graph* of an MDP is a weighted, directed graph whose nodes correspond to the states of the MDP and whose edges correspond to possible transitions between states. Specifically, an edge $(u, v)$ exists on the graph if it is possible to transition from state $u \in \mathcal{S}$ to state $v \in \mathcal{S}$ by taking some action $a \in \mathcal{A}(u)$. In this paper, we use uniform edge weights of 1.

The actions of an MDP take exactly one decision stage to execute. We refer to them as *primitive actions*. Using primitive actions, it is possible to define *abstract actions*, also known as *skills*, whose execution can take a variable number of decision stages. Furthermore, primitive and abstract actions can be combined to form complex action hierarchies. In this work, we represent abstract actions using the options framework [1, 2]. An option $o$ is a three-tuple $(\mathcal{I}_o, \pi_o, \beta_o)$, where $\mathcal{I}_o \subset \mathcal{S}$ is the initiation set, specifying the set of states in which the option can start execution, $\pi_o : \mathcal{S} \times \mathcal{A} \to [0, 1]$ is the option policy, and $\beta_o : \mathcal{S} \to [0, 1]$ is the termination condition, specifying the probability of option termination in a given state. An option policy is ultimately defined in terms of primitive actions—because primitive actions are the fundamental units of interaction between the agent and its environment—but this can be done indirectly by allowing options to call other options, making it possible for agents to act, learn, and plan with hierarchies of primitive and abstract actions.

## 3 Proposed Approach

To define a skill hierarchy, we use modularity maximisation as a central organising principle, applied at multiple time scales. Specifically, we represent the possibilities of interaction between the agent and its environment as a graph and identify partitions of this graph that maximise modularity [3–5].

A *partition* of a graph is a division of its nodes into mutually exclusive groups, called *clusters*. The *modularity* of a partition composed of a set of clusters $C = \{c_1, c_2, \ldots, c_k\}$ is

$$\sum_{i=1}^{k} e_{ii} - \rho a_i^2 \,,$$

where $e_{ii}$ denotes the proportion of total edge weight in the graph that connects two nodes in cluster $c_i$, and $a_i$ denotes the proportion of total edge weight in the graph with at least one end connected to a node in cluster $c_i$. A resolution parameter $\rho > 0$ controls the relative importance of $e_{ii}$ and $a_i$. Intra-cluster edges contribute to both $e_{ii}$ and $a_i$ while inter-cluster edges contribute only to $a_i$. A partition that maximises modularity will have relatively dense connections within its clusters and relatively sparse connections between its clusters.

Finding a partition that maximises modularity for a given graph is NP-complete [6]. Therefore, when working with large graphs, approximation algorithms are needed. The most widely used approximation algorithm is the *Louvain algorithm* [7], which is an agglomerative hierarchical graph clustering approach. While no formal analysis exists, the runtime of the Louvain algorithm has been observed empirically to be linear in the number of graph edges [8]. It has been successfully applied to graphs with millions of nodes and billions of edges [7].

An important feature of the Louvain algorithm is that, as a *hierarchical* graph clustering method, it exposes the structure of a graph at multiple levels of granularity. Specifically, the output of the Louvain algorithm is a sequence of partitions of the input graph. This sequence has a useful structure: multiple clusters found in one partition in the sequence are merged into a single cluster in the next partition in the sequence. In other words, the output is a *hierarchy* of clusters, with earlier partitions containing many smaller clusters that are merged into fewer larger clusters in later partitions. This hierarchical structure forms the basis of our characterisation of a useful multi-level skill hierarchy.

The Louvain algorithm starts by placing each node of the graph in its own cluster. Nodes are then iteratively moved locally, from their current cluster to a neighbouring cluster, until no gain in modularity is possible. This results in a revised partition corresponding to a local maximum of modularity with respect to local node movement. The revised partition is used to define an *aggregate graph* as follows: each cluster in the partition is represented as a single node in the aggregate graph, and a directed edge is added to the aggregate graph if there is at least one edge that connects neighbouring clusters in the corresponding direction. This process is then repeated on the aggregate graph, and then on the next aggregate graph, and so on, until an iteration is reached with no modularity gain. Pseudocode for the Louvain algorithm is presented in Section H of the supplementary material.

Let $h$ denote the number of partitions returned by the Louvain algorithm when applied to the state transition graph. We use each of the $h$ partitions to define a single layer of skills, resulting in an action hierarchy with $h$ levels of abstract actions (skills) above primitive actions. Each level of the hierarchy contains one or more skills for efficiently navigating between neighbouring clusters of the state transition graph. Specifically, we define an option for navigating from a cluster $c_i$ to a neighbouring cluster $c_j$ as follows: the initiation set consists of all states in $c_i$; the option policy efficiently takes the agent from a given state in $c_i$ to a state in $c_j$; the option terminates with probability 1 in states in $c_j$, with probability 0 otherwise.

Taking advantage of the natural hierarchical structure of the partitions produced by the Louvain algorithm, we compose the skills at one level of the hierarchy to define the skills at the next level up. That is, at each level of the hierarchy, option policies call actions (options or primitive actions) from the level below, with primitive actions being called directly by option policies from only the first level of the hierarchy. We call the resulting set of skills the *Louvain skill hierarchy*.

## 4   Related Work

There have been earlier approaches to skill discovery using the state transition graph. The approach we propose here differs from them in two fundamental ways. First, it is novel in its use of modularity maximisation as a central organising principle for skill discovery. Secondly, it produces a multi-level hierarchy, whereas existing graph-based approaches produce hierarchies with only a single level of skills above primitive actions.

Many existing approaches to skill discovery use the state transition graph to identify useful subgoal states and define skills that efficiently take the agent to these subgoals. Suggestions for useful subgoals have often been inspired by the concept of a "bottleneck". They include states that are on the border of strongly-connected regions of the state space [9], states that allow transitions between different regions of the state space [10], and states that lie on the shortest path between many pairs of states [11]. To identify such states, several approaches use graph centrality measures [11–15]. Others use graph

partitioning algorithms [9, 16–22]. Alternatively, it has been proposed that "landmark" states found at the centre of strongly-connected regions of the state-space can be used as subgoals [23].

The proposed approach is most directly related to skill discovery methods that make use of graph partitioning to identify meaningful regions of the state transition graph and define skills for navigating between them [16, 24–28]. Three of these methods use the concept of modularity. One such approach is to generate a series of possible partitions by successively removing the edge with the highest edge betweenness from a graph, then selecting the partition with the highest modularity [25]. A second approach is to generate a partition using the label propagation algorithm and then to merge neighbouring clusters until no gain in modularity is possible [28]. In these two approaches, modularity maximisation is applied after first producing candidate partitions using different criteria. Consequently, the final partition does not maximise modularity overall. The time complexity of the label propagation method is near-linear in the number of graph edges, whereas the edge betweenness method has a time complexity of $O(m^2 n)$ on a graph with $m$ edges and $n$ nodes. A third approach is by Xu et al. [27], who use the Louvain algorithm to find a partition that maximises modularity but, unlike our approach, define skills only for moving between clusters in the highest-level partition, discarding all lower-level partitions. All three methods produce a single level of skills above primitive actions.

Several approaches have used the graph Laplacian [29, 30] to identify skills that are specifically useful for efficiently exploring the state space. It is unclear how to arrange such skills to form multi-level skill hierarchies. In contrast, the proposed approach produces a set of skills that are naturally arranged into a multi-level hierarchy.

While existing graph-based methods do not learn multi-level hierarchies, policy-gradient methods have made some progress towards this goal. Bacon et al. [31] extended policy-gradient theorems [32] to allow the learning of option policies and termination conditions in a hierarchy with a single level of skills above primitive actions. Riemer et al. [33] further generalised these theorems to support multi-level hierarchies. Fox et al. [34] propose an imitation learning method that finds the multi-level skill hierarchy most likely to generate a given set of example trajectories. Levy et al. [35] propose a method for learning multi-level hierarchies of goal-directed policies, with each level of the hierarchy producing a subgoal for the lower levels to navigate towards. However, these methods are not without their limitations. Unlike the approach proposed here, they all require the number of hierarchy levels to be pre-defined instead of finding a suitable number automatically. They do not judiciously define initiation sets, instead making all skills available in all states. They also target different types of problems than we do, such as imitation-learning or goal-directed problems.

## 5 Empirical Analysis

We analyse the Louvain skill hierarchy in the six environments depicted in Figure 1: Rooms, Grid, Maze [23], Office, Taxi [36], and Towers of Hanoi. In all environments, the reward is $-0.001$ for each action and an additional $+1.0$ for reaching a goal state. In Rooms, Grid, Maze, and Office, there are four primitive actions: north, south, east, and west. In Taxi, there are two additional primitive actions: pick-up-passenger and put-down-passenger. Some decisions in Taxi are irreversible. For instance, after picking up the passenger, the agent cannot return to a state where the passenger is still waiting to be picked up. We describe each of these environments fully in Section A of the supplementary material.

When generating partitions of the state transition graph using the Louvain algorithm, we used a resolution parameter of $\rho = 0.05$, unless stated otherwise. When converting the output of the Louvain

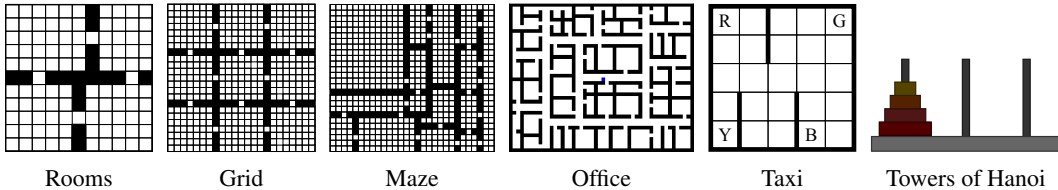

| Rooms | Grid | Maze | Office | Taxi | Towers of Hanoi |

Figure 1: The environments.

algorithm into a concrete skill hierarchy, we discarded all levels of the cluster hierarchy where the mean number of nodes per cluster was less than 4. Our reasoning is that skills that navigate between such small clusters execute for only a very small number of decision stages (often only one or two) and are not meaningfully more abstract than primitive actions. For all methods used in our comparisons, we generated options using the complete state transition graph and learned their policies offline using macro-Q learning [37]. We trained all hierarchical agents using macro-Q learning and intra-option learning [38]. Although these algorithms have not previously been applied to multi-level hierarchies, they both extend naturally to this case. The primitive agent was trained using Q-Learning [39]. The shaded regions on the learning curves represent the standard error in 40 independent runs. All experiments are fully described in Section B of the supplementary material.

Our analysis is directed by the following questions: What is the Louvain skill hierarchy generated in each environment? How does this skill hierarchy impact the learning performance of the agent? How do the results change as the number of states increases? Does arranging skills into a multi-level hierarchy provide benefits over a flat arrangement of the same skills? What is the impact of varying the value of the resolution parameter $\rho$?

**Louvain Skill Hierarchy.** We first examine the cluster hierarchies generated by applying the Louvain algorithm to the state transition graphs of various environments. Figure 2 shows the results in Rooms, Office, Taxi, and Towers of Hanoi. Section D of the supplementary material shows additional results in Grid and Maze.

In Rooms, the hierarchy has four levels. At level three, we see that each room has been placed in its own cluster. Moving up the hierarchy, at level four, two of these rooms have been joined together into a single cluster. Moving down the hierarchy, each room has been divided into smaller clusters at level two, and then into even smaller clusters at level one. The figure illustrates how the corresponding skill hierarchy would enable efficient navigation between and within rooms.

In Office, at the top level, we see six large clusters connected by corridors. As we move lower down the hierarchy, we see that these large clusters have been divided into increasingly smaller regions. At level three, there are many rooms that form their own cluster. At level two, most rooms have been divided into multiple clusters. The figure reveals how the corresponding skill hierarchy would enable efficient navigation of the environment at multiple time scales.

In Taxi, the state transition graph has four disconnected components, each corresponding to one particular passenger destination: R, G, B, or Y. In Figure 2, we show only one of these components, the one where the passenger destination is B. The results for the other components are similar. The Louvain cluster hierarchy has four levels. At the top level, we see four clusters. In three of these clusters, the passenger is waiting at their starting location (R, G, or Y). In the fourth cluster, the passenger is either in-taxi or has been delivered to their destination (B). Navigation between these clusters is unidirectional, with only three possibilities. The three corresponding skills navigate the taxi to the passenger location *and* pick up the passenger. Moving one level down the hierarchy, the clusters produce skills that move the taxi between the left and the right-hand side of the grid, which are connected by the bottleneck state at the centre of the taxi navigation grid.

In Towers of Hanoi, the hierarchy has three levels. At level three, moving between the three clusters corresponds to moving the largest disk to a different pole. Each of these clusters has been divided into three smaller clusters at level two and then into three even smaller clusters at level one. A similar structure exists at levels two and one. The smaller clusters at level two correspond to moving the second-largest disc between different poles; the smallest clusters at level one correspond to moving the third-largest disc between different poles.

In all of these domains, the Louvain skill hierarchy closely matches human intuition. In addition, it is clear how skills at one level of the hierarchy can be composed to produce the skills at the next level.

**Learning Performance.** We compare learning performance with the Louvain skill hierarchy to approaches based on `Edge Betweenness` [25] and `Label Propagation` [28], as well as to the method proposed by `Xu et al.` [27]. These methods are the most directly related to the proposed approach, as discussed in Section 4. In addition, we compare to options that navigate to local maxima of `Node Betweenness` [11], a subgoal-based approach that captures the many different notions of the bottleneck concept in the literature. Similarly to the proposed approach, it is also one of the few approaches in the literature to explicitly characterise a target skill hierarchy for an agent to learn. We also compare to `Eigenoptions` [29], which are derived from the Laplacian of the state transition

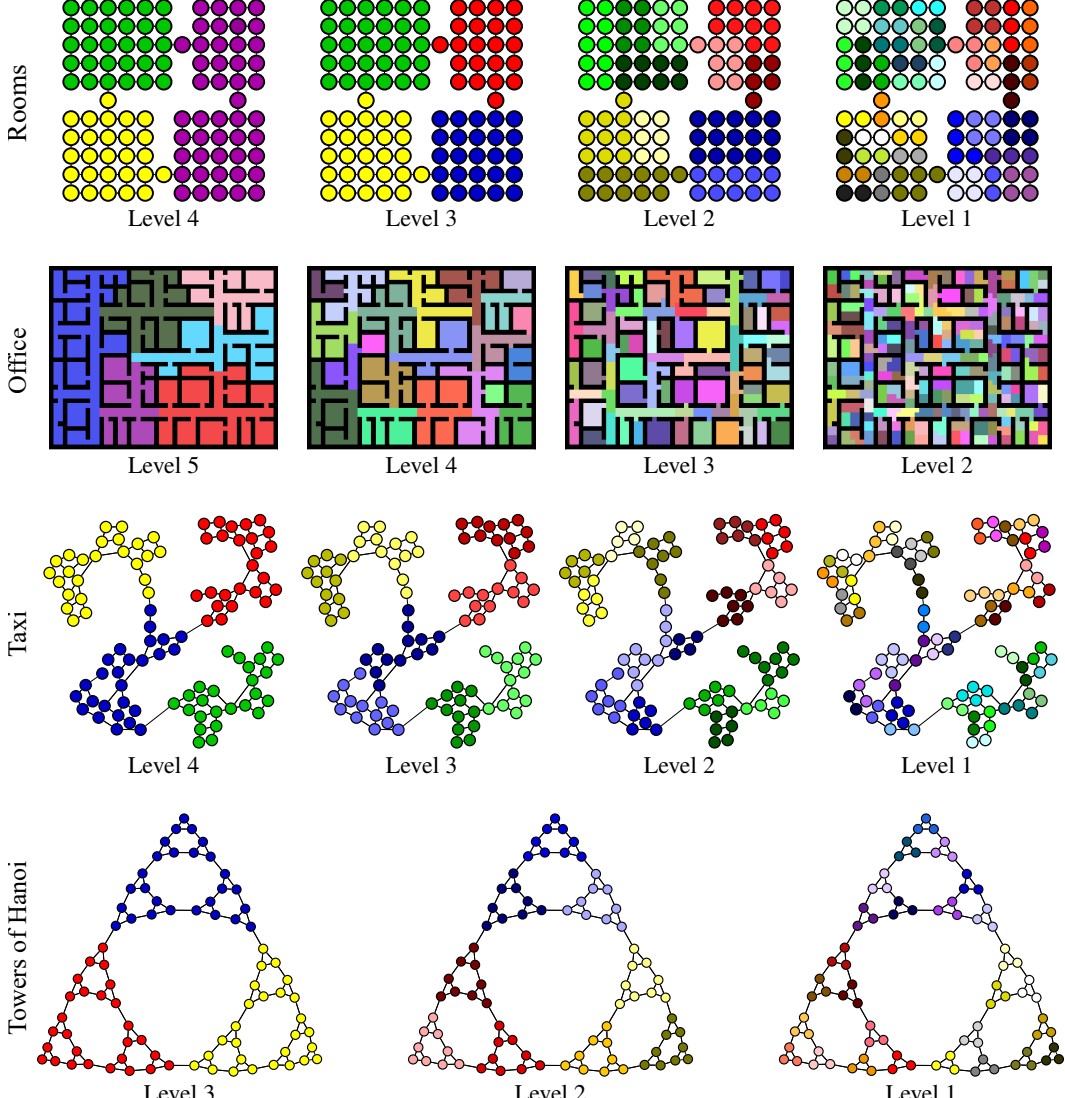

Figure 2: The cluster hierarchies produced by the Louvain algorithm when applied to the state transition graphs representing Rooms, Office, Taxi, and Towers of Hanoi. For Taxi and Towers of Hanoi, the graph layout was determined by using a force-directed algorithm that models nodes as charged particles that repel each other and edges as springs that attract connected nodes.

graph and encourage efficient exploration. Finally, we include a `Primitive` agent that uses only primitive actions. Primitive actions are available to all agents.

We present learning curves in Figure 3. The Louvain agent has a clear and substantial advantage over other approaches in all domains except for Towers of Hanoi, where its performance is much closer to that of the other hierarchical agents, with none of them performing much better than the primitive agent. This is consistent with existing results reported in this domain (e.g., by Jinnai et al. [30]).

Section C of the supplementary material contains further analysis comparing Louvain skills and skills based on node betweenness.

**Scaling to larger domains.** We experimented with a multi-floor version of the Office environment, with floors connected by a central elevator, where two primitive actions move the agent up and down between two adjacent floors. The number of states in this domain can be varied by adjusting parameters such as the number of office floors. We use this environment to explore how the performance of the agent and the Louvain skill hierarchy change as the number of states in the environment increases.

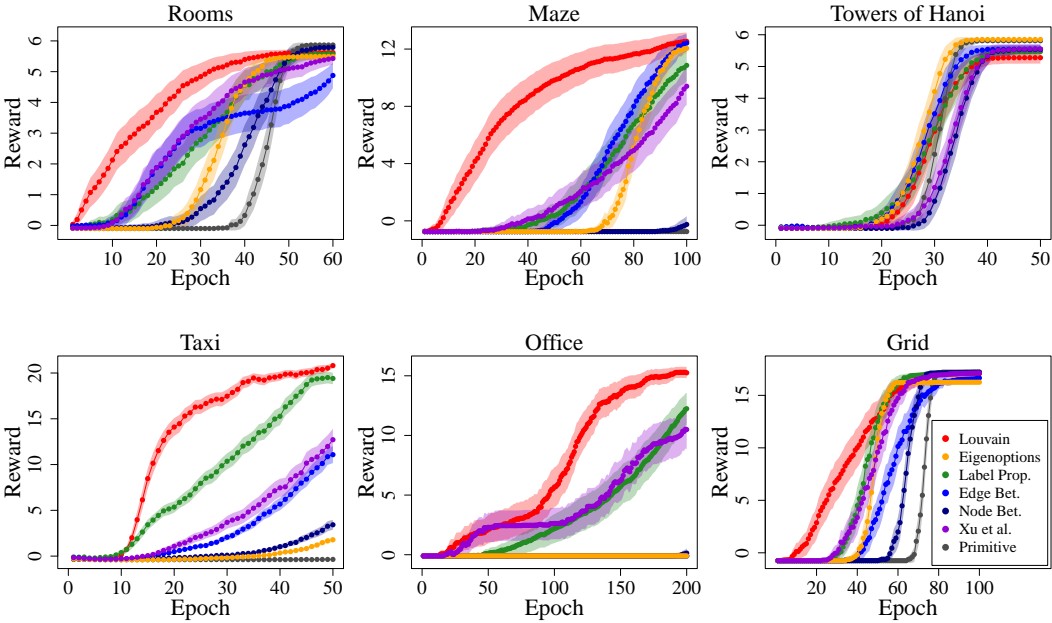

Figure 3: Learning performance. An epoch corresponds to 100 decision stages in Rooms and Towers of Hanoi, 300 in Taxi, 750 in Maze and Grid, and 1000 in Office.

Figure 4a presents results in an Office environment with 2537 states. It shows that the Louvain agent learns much more quickly than all other agents. Some alternatives, including the Eigenoptions agent, fail to achieve any learning.

Figure 4b shows how the Louvain skill hierarchy changes with the size of the state space in the Office environment. The figure shows the depth of the skill hierarchy in a series of fifteen Office environments of increasing size, ranging from a single floor ($\sim 10^3$ states) to one thousand floors ($\sim 10^6$ states). The depth of the hierarchy increased very gradually with the number of states in the environment. The maximum hierarchy depth reached was eight in the largest environment tested.

**Hierarchical versus flat arrangement of skills.** An alternative to the Louvain skill hierarchy is a flat arrangement of the skills, where each skill calls primitive actions directly rather than indirectly through other (lower-level) skills. We expected the hierarchical arrangement to lead to faster learning than the flat arrangement due to the additional learning updates enabled by the hierarchical relationship between the skills. Figure 5 shows that this is indeed the case. In the figure, `Louvain` shows an agent that uses the Louvain skill hierarchy while `Louvain flat` shows an agent that uses the Louvain skills but where the skill policies call primitive actions directly rather than through other skills. In addition, the figure shows a number of agents that use only a single level of the Louvain hierarchy

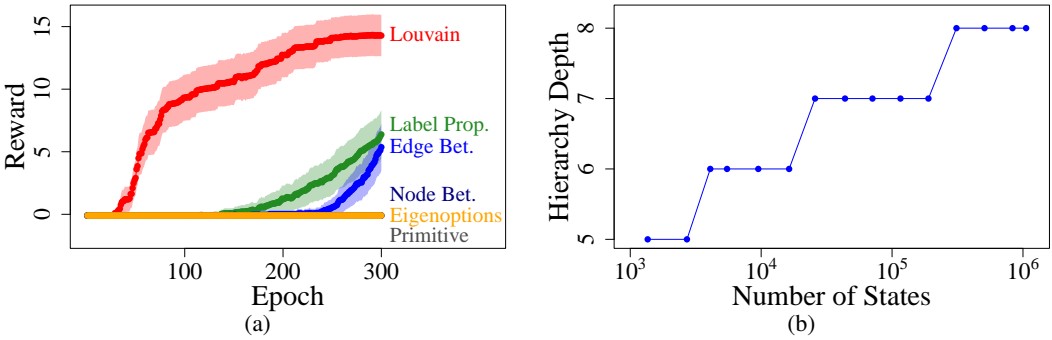

Figure 4: (a) Learning performance in a two-floor Office containing 2537 states. (b) How the depth of the skill hierarchy scales with the size of the state space in multi-floor Office.

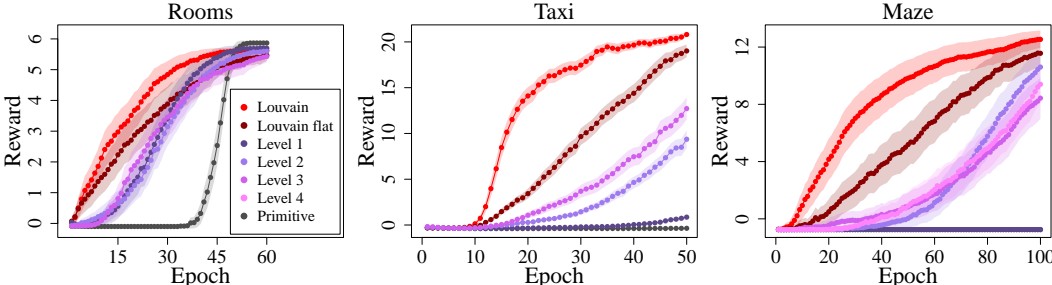

Figure 5: Learning curves comparing various different Louvain agents. An epoch corresponds to 100 decision stages in Rooms, 300 in Taxi, and 750 in Maze. The skill hierarchy contained three levels in Rooms and Taxi, and four levels in Maze.

(`Level 1`, `Level 2`, `Level 3`, or `Level 4`), with option policies that call primitive actions directly. Primitive actions were available to all agents. The figure shows that the hierarchical agent learns more quickly than the flat agent. Furthermore, the agents using individual levels of the Louvain hierarchy learn more quickly than the primitive agent but not as quickly as the agent using the full Louvain hierarchy.

**Impact of the resolution parameter.** When using very high values of the resolution parameter $\rho$, the Louvain algorithm terminates without producing any partitions. On the other hand, when using very low values of $\rho$, it runs until a partition is produced where all nodes are merged into a single cluster.

Figure 6 shows how changing the resolution parameter $\rho$ impacts the Louvain cluster hierarchy in Towers of Hanoi and Rooms. In Towers of Hanoi, at $\rho = 10$, the output is a single level containing many small clusters comprised of three nodes. As $\rho$ gets smaller, the cluster hierarchy remains the same until $\rho = 3.3$, at which point a second level is added. The first level remains identical to the single level produced at $\rho = 10$. The second level contains larger clusters, each formed by merging three of the clusters from the first level. As $\rho$ is reduced further, additional levels are added to the hierarchy. When a new level is added to the hierarchy, the existing levels generally remain the same.

A sensitivity analysis on the value of $\rho$ showed that a wide range of $\rho$ values led to useful skills, and that performance gradually decreased to no worse than that of a primitive agent at higher values of $\rho$. Section E of the supplementary material contains full details of this sensitivity analysis and a more detailed discussion on the choice of $\rho$.

## 6 Discussion and Future Work

An important research direction for future work is incremental learning of Louvain skill hierachies as the agent interacts with its environment—because the state transition graph will not always be available in advance. We explored the feasibility of incremental learning in the Rooms environment.

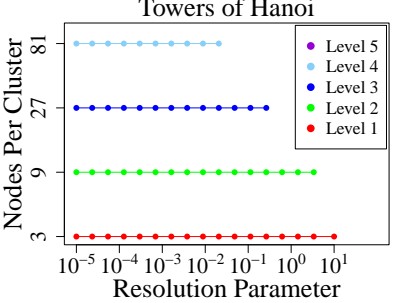 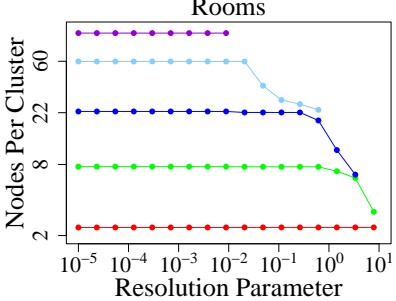

Figure 6: How the Louvain algorithm's output when applied to Towers of Hanoi and Rooms changes with the resolution parameter. The cluster hierarchy had a maximum of four levels in Towers of Hanoi and five levels in Rooms.

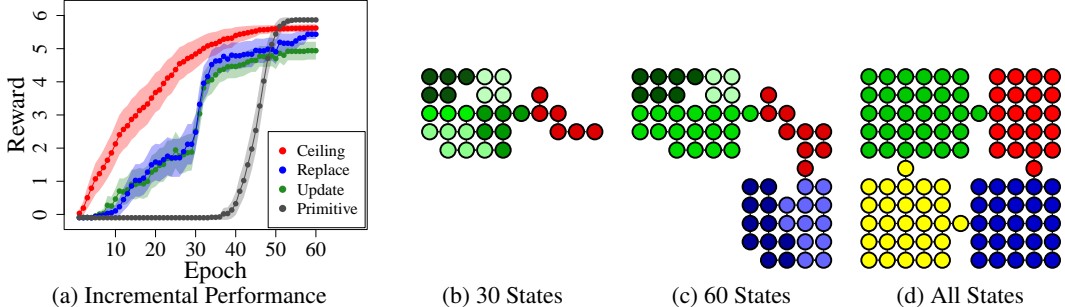

| (a) Incremental Performance | (b) 30 States | (c) 60 States | (d) All States |

Figure 7: (a) Performance when incrementally discovering Louvain options in Rooms. An epoch corresponds to 100 decision stages. (b-d) How the state transition graph and top-level partitions produced by the `Update` approach evolved as an agent explored Rooms. The hierarchy contained 2 levels after visiting 30 states, 3 levels after visiting 60 states, and 5 levels after observing all possible transitions.

The results are shown in Figure 7. The agent started with an empty state transition graph and no skills. Every $m$ decision stages, it updated its state transition graph with new nodes and edges, and it revised its skill hierarchy using one of two approaches: `Replace` or `Update`.

In the first approach (`Replace`), the agent applied the Louvain algorithm to create a new skill hierarchy from scratch. In the second approach (`Update`), the agent incorporated the new information into its existing skill hierarchy, using an approach similar to the Louvain algorithm. This second approach starts by assigning each new node to its own cluster; the new nodes are then iteratively moved locally, between neighbouring clusters (both new and existing) until no modularity gain is possible. This revised partition is used to define an aggregate graph and the entire process is repeated on the aggregate graph, and the next aggregate graph, and so on, until an iteration is reached with no modularity gain. The cluster membership of existing nodes stays fixed; only new nodes have their cluster membership updated. The result of both approaches is a revised set of partitions, from which a revised hierarchy of Louvain skills is derived. Section I of the supplementary material presents pseudocode for the two incremental approaches.

Figure 7a shows the performance of the incremental agents. The agents started with only primitive actions; after decision stages 100, 500, 1000, 3000, and 5000, the state transition graph was updated and the skill hierarchy was revised. The figure compares performance to a `Primitive` agent and an agent using the full Louvain skill hierarchy, whose performance acts as a `Ceiling` for the incremental agents. Both incremental agents learned much faster than the primitive agent and only marginally slower than the fully-informed Louvain agent. The two incremental agents had similar performance throughout training but `Replace` reached a higher level of asymptotic performance than `Update`, as expected. The reason is that partitions produced early in training are based on incomplete information; `Replace` discards these early imperfections while `Update` carries them forward. But there is a trade-off: building a new skill hierarchy from scratch has a higher computational cost than updating an existing one.

Figures 7b–7d show the evolution of the partitions as the `Update` agent performed a random walk in Rooms. The partitions were updated after observing 30 states, 60 states, and all possible transitions. The figure shows that, as more nodes were added, increasingly higher-level skills were produced.

These results demonstrate the feasibility of learning Louvain skills incrementally. A full incremental method for learning Louvain skills may take many forms, and different approaches may be useful under different circumstances, with each having its own strengths and weaknesses. We leave the full development of such algorithms to future work.

Another direction for future work is extending Louvain skills to environments with continuous state spaces such as robotic control tasks. Such domains present a difficulty to all skill discovery methods that use the state transition graph due to the inherently discrete nature of the graph. If the critical step of constructing an appropriate graphical representation of a continuous state space can be achieved, all graph-based methods would benefit. Some approaches have already been proposed in the literature. We use one such approach [40, 20, 26] to examine the Louvain hierarchy in a variant of the Pinball domain [41], which involves manoeuvring a ball around a series of obstacles to reach a goal, as

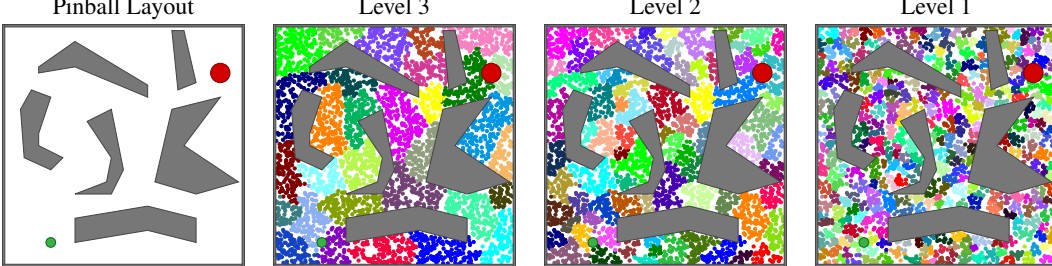

Figure 8: The layout of the Pinball environment—with the green ball in its initial position, the red goal, and several obstacles—and the Louvain cluster hierarchy produced by the Louvain algorithm.

shown in the leftmost panel of Figure 8. The state is represented by two continuous values: the ball's position in the horizontal and vertical directions. At each decision stage, the agent can choose to apply a small force to the ball in each direction. The amount of force applied is stochastic and causes the ball to roll until friction causes it to come to a rest. Collisions between the ball and the obstacles are elastic.

We sampled $4000$ states, added them to the state transition graph, then added an edge between each node and its $k$-nearest neighbours according to Euclidean distance, assigning an edge weight of $e^{-4d^2}$ to an edge that connects two locations $u$ and $v$ with Euclidean distance $d$ between them. We then applied the Louvain algorithm to the resulting graph. The result was the cluster hierarchy shown in Figure 8. It had three levels. Moving between clusters in the top level corresponds to skills that enable high-level navigation of the state space, and take into account features such as the natural bottlenecks caused by the obstacles, allowing the agent to efficiently change its position. Moving between clusters in the lower-level partitions enable more local navigation.

Currently, Louvain skills at one level of the hierarchy are composed of skills from only the previous level. Such skills may not provide the most efficient navigation between clusters. Future work could consider composing skills from all lower levels, including primitive actions.

Because Louvain skills are derived from the connectivity of the state transition graph, we expect them to be suitable for solving a range of problems in a given environment. Examining their use for transfer learning is a useful direction for future work.

A possible difficulty with building multi-level skill hierarchies is that having a large number of skills can end up hurting performance by increasing the branching factor. One solution that has been explored in the context of two-level hierarchies is pruning less useful skills from the hierarchy [21, 42]. Further research is needed to address this potential difficulty.

A key open problem for graph-based skill discovery methods generally is how to best make use of state abstraction. We suggest that the most natural place to introduce state abstraction is when constructing the state transition graph itself. Instead of a concrete state, each node could represent an abstract state based on some learned representation of the environment. The proposed method—and any other existing graph-based method—could then directly use this abstract state transition graph to define a skill hierarchy.

Lastly, we note that the various characterisations of useful skills proposed in the literature, including the one proposed here, are not necessarily competitors. To solve complex tasks, it is likely that an agent will need to use many different types of skills. An important avenue for future work is exploring how different ideas on skills discovery can be used together to enable agents that can autonomously develop complex and varied skill hierarchies.

## Acknowledgments and Disclosure of Funding

We would like to thank the members of the Bath Reinforcement Learning Laboratory for their constructive feedback. This research was supported by the Engineering and Physical Sciences Research Council [EP/R513155/1] and the University of Bath. This research made use of Hex, the GPU Cloud in the Department of Computer Science at the University of Bath.

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
