# Creating Multi-Level Skill Hierarchies in Reinforcement Learning

## SUPPLEMENTARY MATERIAL

**Joshua B. Evans**
Department of Computer Science
University of Bath
Bath, United Kingdom
jbe25@bath.ac.uk

**Özgür Şimşek**
Department of Computer Science
University of Bath
Bath, United Kingdom
o.simsek@bath.ac.uk

## A  Environments

**Gridworlds** include Rooms, Grid, Office, and Maze [1]. They had four primitive actions: north, south, east, and west. These actions move the agent in the intended direction (unless the intended direction faces a wall, in which case the agent remains in the same state). The reward is $-0.001$ for each action and an additional $+1.0$ for reaching the goal state. There is a single start state and a single goal state, selected for each run from a list of possibilities.

**Multi-Floor Office** is an extension of Office to multiple floors. All foors are connected by an elevator, which occupies the same grid square on each floor, where the agent has two additional primitive actions: up and down. These actions move the agent to the adjacent floor in the intended direction (unless there is no other floor in that direction).

**Taxi** is a $5 \times 5$ grid with four special squares (R, G, B, and Y) that serve as possible pick-up and drop-off locations for a passenger. An episode starts with the taxi at a random square, the passenger at a random special square, and the destination another random special square. Six actions are available in each state: north, south, east, west, pick-up, and put-down. The navigation actions are identical to those in the gridworlds, as described above. Pick-up and put-down have the intended effect when appropriate; otherwise they do not change the state. The reward is $-0.001$ for each action and an additional $+1.0$ when the passenger is put down at the destination.

**Towers of Hanoi** contains four discs of different sizes, placed on three poles. An episode starts with all discs stacked on the leftmost pole. Primitive actions move the top disc from one pole to any other pole, with the constraint that a disc cannot be placed on top of a disc smaller than itself. The reward is $-0.001$ for each action and an additional reward of $+1.0$ at the goal state, where when all three discs are stacked on the rightmost pole.

## B  Methodology

**Generating options.** To generate Louvain options, the Louvain algorithm ($\rho = 0.05$) was applied to the state transition graph, resulting in a set of partitions. Any partition where the mean number of nodes per cluster was smaller than $4$ was discarded. For all remaining partitions, options were defined for efficiently taking the agent from a cluster $c_i$ to each of its neighbouring clusters $c_j$ if a directed edge existed from a node in $c_i$ to a node in $c_j$. The Louvain options were arranged into a multi-level hierarchy, where options for navigating between clusters in partition $i$ could call skills for navigating between clusters in partition $i - 1$. Only options at the lowest level of the hierarchy could call primitive actions. Options generated using alternative methods called primitive actions directly.

37th Conference on Neural Information Processing Systems (NeurIPS 2023).

Eigenoptions [2] were generated by computing the normalised Laplacian of the state transition graph and using its eigenvectors to define pseudo-reward functions for each Eigenoption to maximise. In Office, the first 32 eigenvectors (and their negations) were used. In all other environments, the first 16 eigenvectors (and their negations) were used. Options for navigating to local maxima of betweenness [3] were generated by selecting all local maxima as subgoals and defining options for navigating to each subgoal from the nearest 30 states.

**Learning option policies.** For all methods except Eigenoptions, option policies were learned using macro-Q learning [4], with learning rate $\alpha = 0.6$, initial action values $Q_0 = 0$, and discount factor $\gamma = 1$. All agents used an $\epsilon$-greedy exploration strategy with $\epsilon = 0.2$. For options based on clustering, the agent started in a random state in the source cluster. It received a reward of $-0.01$ for each action taken and an additional $+1.0$ for reaching a state in the goal cluster. For options based on node betweenness, the agent started in a random state in the initiation set. It received a reward of $-0.01$ at each decision stage, an additional $+1.0$ for reaching the subgoal state, and an additional $-1.0$ for leaving the initiation set. For Eigenoptions, value iteration was used to produce policies that maximised the pseudo-reward function of each Eigenoption.

**Producing learning curves.** All agents were trained using macro-Q [4] and intra-option learning [5] updates, which were performed every time an option at any level of the hierarchy terminated. A learning rate of $\alpha = 0.4$, discount factor of $\gamma = 1$, and initial action values of $Q_0 = 0$ were used in all experiments. All agents used an $\epsilon$-greedy exploration strategy with $\epsilon = 0.1$. All learning curves show evaluation performance. After training each agent for one epoch, the learned policy was evaluated (with exploration and learning disabled) in a separate instance of the environment.

**Applying the Louvain algorithm in Pinball.** The state transition graph was constructed by following the method used by Mahadevan and Maggioni [6]. Initially, 4000 states were randomly sampled and a node representing each state was added to the graph. Edges were then added between each node and its 10 nearest neighbours. An between two locations $u$ and $v$ with Euclidean distance $d$ between them was assigned a weight of $e^{-4d^2}$. Finally, the Louvain algorithm ($\rho = 0.05$) was applied to the resulting graph.

## C   Comparison to Skills that Navigate to Local Maxima of Betweenness

Here we explore the relationship between Louvain skills and the skill chracterisation proposed by Şimşek and Barto [3]. The latter is a subgoal-based approach that captures various definitions of the bottleneck concept. It defines skills that navigate to local maxima of betweenness. Both approaches address the conceptual problem of what makes a useful skill, explicitly defining a target set of skills for the agent to learn.

There is substantial overlap between Louvain skills and skills that navigate to local maxima of betweenness. They both include skills that traverse bottleneck states, including those that navigate between rooms in Rooms, picking up the passenger in Taxi, and navigating different parts of the grid in Taxi. In Towers of Hanoi, all Louvain skills traverse states that are also identified as local maxima of betweenness. The highest local maxima of betweenness correspond to the states that separate Louvain clusters at level 3; the second highest local maxima of betweenness correspond to the states that separate Louvain clusters at level 2.

On the other hand, there are many Louvain skills that do not correspond to navgating to local maxima of betweenness. Examples include the Louvain skills that navigate within a single room in Rooms.

Most importantly, Louvain skills and skills that navigate to local maxima of betweenness differ in how they can be arranged hierarchically. Even in Towers of Hanoi, where there is a clear hierarchical structure between the larger and smaller local maxima of betweenness, it is not clear how to exploit the betweenness metric to form a multi-level hierarchy. In contrast, the modularity metric approximated by the Louvain algorithm provides a clear and principled way of building a multi-level hierarchy.

## D   Cluster Hierarchies

Figure 1 shows the cluster hierarchies produced by the Louvain algorithm when applied to the state transition graphs of Grid and Maze. It also shows the first level of the cluster hierarchy in Office, which was omitted in the main paper due to space limitations.

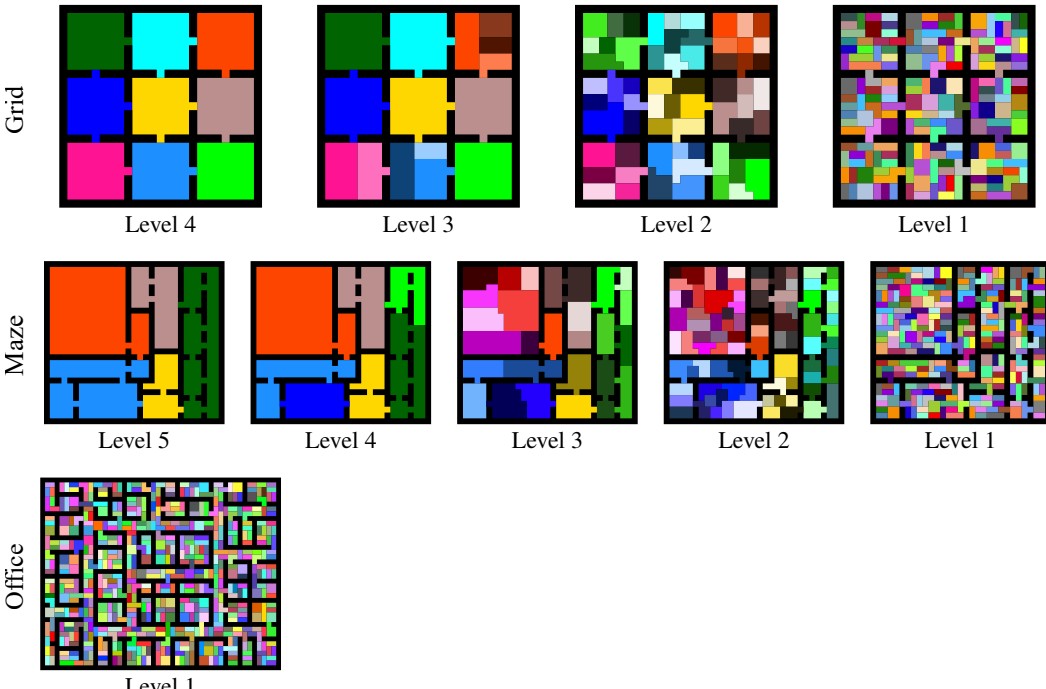

Figure 1: Top two rows: Cluster hierarchies produced by the Louvain algorithm in Grid and Maze. Bottom row: The lowest level of the cluster hierarchy in Office.

# E    Sensitivity to the Resolution Parameter

Figure 2 shows the performance of agents in Rooms and Towers of Hanoi using Louvain skills generated using different values of $\rho$. Louvain skills created using lower values of $\rho$ consistently outperformed those created by using higher values, and very high values ($\rho \geq 10$) generally led to performance similar to that obtained by using primitive actions only. Lower $\rho$ values lead to deeper hierarchies that contain skills for navigating the environment over varying timescales. In contrast, higher $\rho$ values result in shallower skill hierarchies that contain few—or, in the extreme, no—levels of skills above primitive actions. While there are better and worse values of $\rho$, it may be argued that there is no "bad" choice; lowering $\rho$ will result in deeper hierarchies, but existing levels of the hierarchy will remain intact.

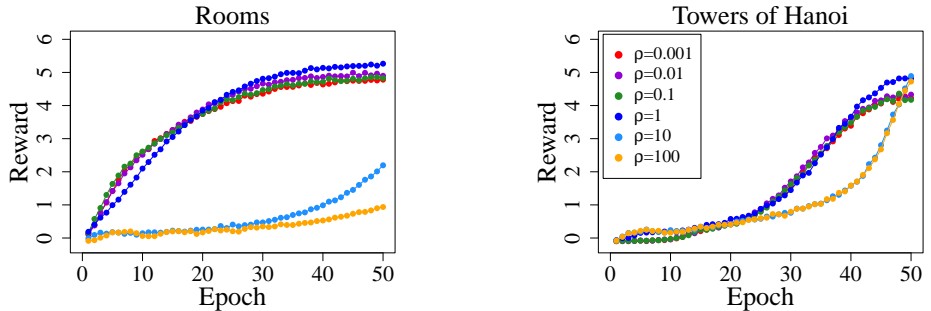

Figure 2: Agent performance with Louvain skills generated using different settings of the resolution parameter $\rho$.

## F   Compute Resource Usage

The experiments were run using a shared internal CPU cluster with the specifications shown below. Approximately 40 CPU cores were utilised for approximately 336 hours when producing the final set of results presented in the paper. Prior to this, approximately 20 CPU cores were utilised for approximately 168 hours during preliminary testing. GPU acceleration was not used because the experiments involved tabular reinforcement learning methods.

| | |
|---|---|
| **Processor** | $2\times$ AMD EPYC 7443 |
| **Cores per Processor** | 24 Cores |
| **Clock Speed** | 2.85GHz–4GHz |
| **RAM** | 512 GB |
| **RAM Speed** | 3200MHz DDR4 |

## G   Source Code

An implementation of the proposed approach, implementations of the test environments, and the code used to generate all results presented in the paper can be found in the following GitHub repository: https://github.com/bath-reinforcement-learning-lab/Louvain-Skills-NeurIPS-2023

# H  Louvain Algorithm

Algorithm 1 shows pseudocode for the Louvain algorithm [7]. It takes a graph $G_0 = (V_0, E_0)$ as input and outputs a set of partitions of that graph into clusters. Each iteration of the algorithm (lines 4–25) produces one partition of the graph.

---

**Algorithm 1:** Louvain Algorithm

---

1 **parameters:** resolution parameter $\rho \in \mathbb{R}^+$
2 **input:** $G_0 = (V_0, E_0)$        `// e.g., the state transition graph of an MDP`
3 $i \leftarrow 0$
4 **repeat**
5    $C_i \leftarrow \{\{u\} \mid u \in V_i\}$        `// define singleton partition`
6    $Q_{\text{old}} \leftarrow$ modularity from dividing $G_i$ into partition $C_i$
7    **repeat**
8      $C_{\text{before}} \leftarrow C_i$
9      **foreach** $u \in V_i$ **do**
10        find clusters neighbouring $u$, $N_u \leftarrow \{c \mid c \in C_i, v \in V_i, v \in c, (u,v) \in E_i\}$
11        compute the modularity gain from moving $u$ into each cluster $c \in N_u$
12        update $C_i$ by inserting $u$ into cluster $c \in N_u$ that maximises modularity gain
13      **end foreach**
14      $C_{\text{after}} \leftarrow C_i$
15    **until** $C_{before} = C_{after}$       `// no nodes changed clusters during an iteration`
16    $Q_{\text{new}} \leftarrow$ modularity from dividing $G_i$ into revised partition $C_i$
17    **if** $Q_{new} > Q_{old}$ **then**
18      $V_{i+1} \leftarrow \{c \mid c \in C_i\}$
19      $E_{i+1} \leftarrow \{(c_j, c_k) \mid c_j \in C_i, c_k \in C_i, (u,v) \in E_i, u \in c_j, v \in c_k\}$
20      $G_{i+1} \leftarrow (V_{i+1}, E_{i+1})$    `// derive aggregate graph from current partition`
21      $i \leftarrow i + 1$
22    **else**
23      break
24    **end if**
25 **end**
26 **output:** partitions $C_0, \ldots, C_{i-1}$

---

# I Incremental Discovery of Louvain Skills

Algorithm 2 presents an approach for incrementally developing a hierarchy of Louvain skills over time, starting with only primitive actions. To update the agent's skill hierarchy, Algorithm 2 calls upon either Algorithm 1 or Algorithm 3, depending on whether the agent's existing skill hierarchy is to be replaced or updated. Algorithm 3 presents an approach for incrementally updating Louvain partitions. It integrates new nodes into an existing cluster hierarchy.

---

**Algorithm 2:** Incremental Discovery of Louvain Skills

---

1 **input:** variant $\in \{1, 2\}$     `// which variant of the incremental algorithm to use`
2 **input:** $N = \{n_1, n_2, \ldots\}$     `// decision stages to revise skill hierarchy after`
3 $V \leftarrow \emptyset$                                   `// initialise empty set of nodes`
4 $E \leftarrow \emptyset$                                   `// initialise empty set of edges`
5 $G \leftarrow (V, E)$                  `// initialise empty state transition graph (STG)`
6 $C \leftarrow \emptyset$                  `// initialise empty set of partitions of the STG`
7 $V_{\text{new}} \leftarrow \emptyset$               `// initialise empty set for recording novel states`
8 $E_{\text{new}} \leftarrow \emptyset$            `// initialise empty set for recording novel transitions`
9 initialise $Q(s, a)$ for all $s \in \mathcal{S}, a \in \mathcal{A}(s)$ arbitrarily, with $Q(\text{terminal state}, \cdot) = 0$
10 $t \leftarrow 0$
11 **repeat**
12     initialise environment to state $S$
13     **if** $S \notin V$ **then**
14        $V_{\text{new}} \leftarrow V_{\text{new}} \cup \{S\}$
15     **end if**
16     **while** $S$ *is not terminal* **do**
17        choose $A$ from $S$ using policy derived from $Q$ (e.g., $\epsilon$-greedy)
18        take action $A$, observe next-state $S'$ and reward $R$
19        perform macro-Q and intra-option updates using $(S, A, S', R)$
20        $S \leftarrow S'$
21        **if** $S' \notin V$ **then**
22           $V_{\text{new}} \leftarrow V_{\text{new}} \cup \{S'\}$          `// add novel state to set of new nodes`
23        **end if**
24        **if** $(S, S') \notin E$ **then**
25           $E_{\text{new}} \leftarrow E_{\text{new}} \cup \{(S, S')\}$   `// add novel transition to set of new edges`
26        **end if**
27        **if** $t \in N$ **then**
28           add each state $u \in V_{\text{new}}$ to $V$
29           add each transition $(u, v) \in E_{\text{new}}$ to $E$
30           $V_{\text{new}} \leftarrow \emptyset$
31           $E_{\text{new}} \leftarrow \emptyset$
32           **if** *variant* $= 1$ **then**
33              $C \leftarrow$ partitions of the STG derived from $(V, E)$ using Algorithm 1
34              *replace* existing skill hierarchy with skills derived from $C$
35           **else if** *variant* $= 2$ **then**
36              $C \leftarrow$ partitions of the STG derived from $(V, E, C)$ using Algorithm 3
37              *revise* existing skill hierarchy based on skills derived from $C$
38           **end if**
39           initialise entries in $Q$ for all new skills arbitrarily, with $Q(\text{terminal state}, \cdot) = 0$
40           remove entries from $Q$ for all skills that no longer exist in the revised hierarchy
41        **end if**
42        $t \leftarrow t + 1$
43     **end while**
44 **end**

---

**Algorithm 3:** Update Louvain Partitions

---

1  **parameters:** resolution parameter $\rho \in \mathbb{R}^+$
2  **input:** $G_0 = (V_0, E_0)$     `// e.g., the state transition graph (STG) of an MDP`
3  **input:** $C = \{C_0, C_1, \ldots, C_n\}$     `// an existing set of partitions of the STG`
4  $i \leftarrow 0$
5  **repeat**
6     $V_{\text{new}} \leftarrow$ nodes in $V_i$ not assigned to any cluster in $C_i$
7     $C_i \leftarrow C_i \cup \{\{u\} \mid u \in V_{\text{new}}\}$     `// define singleton partition over new nodes`
8     $Q_{\text{old}} \leftarrow$ modularity from dividing $G_i$ into partition $C_i$
9     **repeat**
10         $C_{\text{before}} \leftarrow C_i$
11         **foreach** $u \in V_{new}$ **do**
12           find clusters neighbouring $u$, $N_u \leftarrow \{c \mid c \in C_i, v \in V_i, v \in c, (u,v) \in E_i\}$
13           compute the modularity gain from moving $u$ into each cluster $c \in N_u$
14           update $C_i$ by inserting $u$ into cluster $c \in N_u$ that maximises modularity gain
15         **end foreach**
16         $C_{\text{after}} \leftarrow C_i$
17     **until** $C_{before} = C_{after}$     `// no nodes changed clusters during an iteration`
18     $Q_{\text{new}} \leftarrow$ modularity from dividing $G_i$ into revised partition $C_i$
19     **if** $Q_{new} > Q_{old}$ *or* $i < |C|$ **then**
20         $V_{i+1} \leftarrow \{c \mid c \in C_i\}$
21         $E_{i+1} \leftarrow \{(c_j, c_k) \mid c_j \in C_i, c_k \in C_i, (u,v) \in E_i, u \in c_j, v \in c_k\}$
22         $G_{i+1} \leftarrow (V_{i+1}, E_{i+1})$     `// derive aggregate graph from current partition`
23         $i \leftarrow i + 1$
24     **else**
25         break
26     **end if**
27  **end**
28  **output:** partitions $C_0, \ldots, C_{i-1}$

---