# OpenReview forum: "Creating Multi-Level Skill Hierarchies in Reinforcement Learning"
_NeurIPS.cc/2023/Conference — NeurIPS 2023 poster_

### Official Review · Reviewer_UjdQ · 2023-07-03

**Soundness:** 1 poor
**Presentation:** 2 fair
**Contribution:** 1 poor
**Rating:** 2
**Confidence:** 4

**Summary:**

The paper proposes a method to discover skill hierarchy by applying hierarchical graph clustering methods to expose the structure. The proposed method introduces Louvain method for skill hierarchy revealing and produces skill hierarchy with multiple-grained of action abstractions. The proposed method is tested on six different environments and compared with other methods.

**Strengths:**

This paper introduces the concept of graph clustering to discover action hierarchies, which is an interesting idea. The references are described in details, which makes the intuition and technique of the proposed method more reasonable. It is appreciated also the experiments in six different problems and the evaluation with other methods, as well as the ablation study.

**Weaknesses:**

There are several weaknesses. First, this paper is poorly written. There are a lot of grammatical errors and unprofessional writing styles, which makes it difficult to understand. Technical writing in Proposed Approach part in this paper is also limited, causing a great deal of confusion about its proposed method.

This paper attempts to solve the problem of action hierarchy discovery of agents. This problem is extremely difficult and could not be well solved by introducing Louvain-based graph clustering methods as described in the paper. More details will be described in the limitations. Moreover, the proposed approach section in the paper spends plenty of words to describe the process of the Louvain algorithm. However, the Louvain algorithm has been proposed in 2008 and there have been a lot of improved works during 2010-2015, which should not be the focus of the paper.

In addition, although there are many tasks in the experiment section, these problems are all simple and some of them can be solved well even using simple planning methods. It is unnecessary to discover action hierarchies in these scenarios. Agents’ skills should be learned in more complex scenarios with more constraints, such as StarCraft II and the Minecraft (the recently popular environment), but these complex scenarios are not considered in this paper.


**Questions:**

How about the learning efficiency of the proposed algorithm? Does the size of the graph limit the learning efficiency of the proposed method compared to the original reinforcement learning algorithm?

I am confused that there is no diversity in the skills formed by combining only 4 actions with same type in these simple scenarios including Rooms. Poor diversity limited the performance of policies learned from the reinforcement learning framework.

There are too many confusing descriptions, including:

•	"an autonomous agent" in Line1

•	"graphical structure" in Line2

•	"its environment" in Line2

•	"multiple levels of abstraction" in Line4

•	"regions of the state" in Line5

•	"connected within themselves" in Line6

•	"various levels of granularity" in Line16

•	"outcome" in Line17

•	"characterisation" in Line19

•	"existing approaches to skill discovery" in Line55

•	"are not without their limitations" in Line72


**Limitations:**

Existing reinforcement learning algorithms based on the state-action space have achieved great results, especially in simple scenarios, which can achieve efficient exploration in the policy space. The reason for introducing skills to split the primitive action space into multiple parts for separate exploration is that the state action search space is so large in facing with more complex scenarios that the primitive reinforcement learning algorithm is inefficient for exploration. However, it is difficult for the skill-based approach to converge to the global optimal solution. The possibility of convergence to a local optimal solution in each subspace is high, which is more likely to occur in simple scenarios. This paper does not propose a theoretical guarantee on the learning efficiency of the algorithm and lacks the optimization analysis of the algorithm. In addition, it does not conduct experimental evaluation on complex scenarios to answer aforementioned questions.

Louvain is a commonly used in community discovery to mine graph clusters. However, the learning process of Louvain algorithm and the learning process of reinforcement learning framework are independent. How can the clustering information of state action space mined by Louvain algorithm actively guide the reinforcement learning? How can the policy search process in the state-action space be fed back to the mining process of Louvain? I think it is difficult. In addition, the modularity seems to be independent of the reward in reinforcement learning, which may lead to the inability of combining the Louvain algorithm with the reinforcement learning algorithm, no matter what approach is taken.

I wonder if graph partitioning can really guide skill learning. First concerning about the model-based reinforcement learning, which generates a complete state action transition graph. However, once the transition graph is completely knowable, traditional reinforcement learning methods can achieve great performance. There is no need to develop reinforcement learning method based on graph partitioning. In case of model-free reinforcement learning, where the policy learning of traditional reinforcement learning methods is difficult, graph partitioning will play a positive role. However, if the transition graph is unknown, it is impossible to reveal expective clusters of the transition graph. Therefore, the proposed method in this paper would be ineffective. But I think that combining graph partitioning method with model-free reinforcement learning with incomplete transition graph is a promising direction.

---

> ### Author Rebuttal · Authors · 2023-08-09
>
> Thank you for the time and attention you have given to our paper.
>
> LA: Louvain Algorithm
>
> **Q1: How about the learning efficiency of the proposed algorithm? Does the size of the graph limit the learning efficiency of the proposed method compared to the original reinforcement learning algorithm?**
>
> We see consistently, across domains, that the learning efficiency increases when Louvain skills are available to the agent (Figure 3). We also see that the learning efficiency gained from Louvain skills (compared to primitive actions) increases with the size of the domain: the largest performance gap between the Louvain agent and the primitive agent is in the largest domain we tested (Figure 5c).
>
> **Q2: I am confused that there is no diversity in the skills formed by combining only 4 actions with same type in these simple scenarios including Rooms.**
>
> It is possible we do not understand what you mean by diversity because we are confused by this question. There is in fact significant diversity in the Louvain skills.
>
> For example, consider the *Level 3* skills in Rooms (see Figure 2, top row). There are 8 skills at this level, two from each room, and each of these skills takes the agent in a different direction. For instance, in the upper-left room (depicted in green), two skills can be initiated: one (efficiently) takes the agent to the upper-right room (red) and the other to the lower-left room (yellow). And this is only the diversity at level 3 of the skill hierarchy. Skills at other levels of the hierarchy are similarly diverse but, *in addition*, they introduce diversity in the skills’ reach (trajectory length; the degree of temporal abstraction). For instance, Level 2 skills are shorter (in trajectory length), and Level 1 skills are even shorter, than Level 3 skills.
>
> In summary, we see diversity in skill initiation sets, skill policies, and trajectory lengths. Furthermore, collectively, the skills cover the state space very well.
>
> **Q3: There are too many confusing descriptions (...)**
>
> We are sorry to hear this and would very much like to remedy it. At the same time, we are unsure what is confusing about the words and phrases you have highlighted. Many, such as *autonomous agent* and *environment*, are fundamental concepts in reinforcement learning. Others, such as *graphical structure* are clearly illustrated in the paper. We would welcome any concrete comments that would help us improve the writing.
>
> **[Skill discovery] is extremely difficult and could not be well solved by introducing Louvain-based graph clustering methods as described in the paper.**
>
> We agree, skill discovery is very difficult. This is exactly why it is important to explore a wide variety of approaches and to not prematurely discard any of them, especially ones that show promise in small-scale problems, such as the one we propose here.
>
> We strongly oppose the statement that skill discovery cannot be solved by introducing Louvain-based graph clustering methods. There is no basis for this statement. Skill discovery is an open problem and Louvain-based graph clustering methods may well be part of the solution.
>
> **The Louvain algorithm has been proposed in 2008 and there have been a lot of improved works during 2010-2015 (...) [it] should not be the focus of the paper.**
>
> LA remains one of the most popular and highly-performing modularity optimisation algorithms. Although several improvements have been proposed to its original formulation, most of these modifications improve the algorithm’s runtime but do not impact its final output. Those modifications that do modify its output often result in the hierarchical structure being harder to extract, and this makes them unsuitable for our purposes.
>
> We explain LA in the paper because otherwise the skill hierarchy it generates would not be as clear to the reader.
>
> **Agents’ skills should be learned in more complex scenarios with more constraints, such as StarCraft II and the Minecraft (...)**
>
> This is an unrealistically high bar. Adopting it would be detrimental to research progress on this very important subject, and this is recognised by the research community: Papers on skill discovery are continuously being published at NeurIPS and other high-profile venues, such as ICML; we are not aware of a single paper that meets this bar. In fact, many new ideas are presented and tested in small, discrete domains. e.g., Bar, Talmon, & Meir (ICML 2020), Jinnai et al. (ICML 2019).
>
> **Finally, we would like to respond to the 3 limitations raised by the reviewer.**
>
> 1a. Factual correction: skill-based approaches do converge to the global optimal solution. If primitive actions are included in an agent’s action set along with skills (as we have done), then existing convergence guarantees from the primitive-only case continue to hold. Some HRL frameworks might lose global convergence (e.g., MAX-Q) but not the options framework (see Precup, Sutton & Singh, Theoretical results on RL with temporally abstract options, ECML, 1998).
>
> 1b. Theoretically analysing the learning efficiency of agents using a specific skill hierarchy is difficult: such analysis is not present even in the most well-known skill discovery papers. We argue that this is not a reasonable requirement from the paper.
>
> 2a. The relationship between LA and RL is simple: LA provides input to RL.
>
> 2b. Clustering by LA does indeed guide RL: it impacts exploration; it determines what the skills are and skills are part of the policy learned by RL.
>
> 2c. RL does not inform LA, but why is this a weakness?
>
> 2d. You may find it useful to see the discussion of reward at the end of our rebuttal to Reviewer cG6b.
>
> 3. The paper should not be evaluated as a skill discovery algorithm. We are not presenting a skill discovery algorithm but a hypothesis on what makes a useful skill hierarchy. To evaluate this hypothesis, we necessarily use the transition graph. We took great care in writing the paper to make this point clearly and repeatedly.

---

> > ### Comment · Reviewer_UjdQ · 2023-08-18
> >
> > Thank you for the comments. But I am keeping my original score given the limitations of your work:
> > 1. The paper claims that the main contribution is a characterisation of a useful action hierarchy which is an action structure. The action structure is intuitive and easily constructed. Many works have made efforts in this aspect [1-7].  The action structure can even be represented as a graph structure, and many works have done so [8-10]. Therefore, it should not be considered as a contribution of the paper. The paper also claims that it uses Louvain-based graphical representation method for the characterisation of the action hierarchy. However, the Louvain algorithm has long been proposed with many improvement works[11-15]. Besides, as mentioned in the related work, many works have used graph partitioning methods for skill discovery[16-19]. Therefore, the real contribution of this paper is only constructing the state-action space for option learning based on the graph partitioning results. However, the option learning method is not described either in this paper or in the supplementary material.  From the description of the Analysis sector, it seems that this method is not different from the classical reinforcement learning methods.
> > **In summary, what this paper does is merely using the Louvain method to generate clustering results of the transition graph for training classical RL algorithms, without substantial innovation.**
> > 2. In the Introduction section of this paper, the author claims that proposed approach differs from existing methods in two ways: parition the interaction graph by maximising modularity and producing a multi-level hierarchy. However, the first difference has already been clearly proposed in the fast unfolding method[11].  Part of the second, which generates a hierarchical clustering structure has also been described in the fast unfolding method[11]. Therefore, the only real difference between this method and existing work isonly a very small part mentioned in the paper, which is constructing the state-action space for option learning based on the clustering results. In fact, the proposed method does not have much novelty compared to classical reinforcement learning methods. **Therefore,  there is so little original content in this paper that it would not be accepted in a top-tier conference**.
> > 3. The author notes that there is no basis for this statement that skill discovery cannot be solved by introducing Louvain-based graph clustering methods and skill discovery is an open problem and Louvain-based graph clustering methods may well be part of the solution. **I wonder to know if there's any basis for the author's claim?** The open problem cannot be solved by mere textual description. We consider two commonly used methods.
> > - The first is the theoretical analysis. Is there any theoretical analysis in this paper? **Neither the main content nor the supplementary material contains the theoretical analysis of the proposed approach.**
> > - The second is experiments. Although the paper describes several experimental scenarios, these scenarios are all too simple. These scenarios were usually used in papers published 20 years ago, but recent work has rarely used such simple scenarios for experiments, except for papers about the theoretical analysis of reinforcement learning. Recent work has typically used scenarios such as Atari and Mujoco for experiments. As stated in the previous part, there is no theoretical analysis in this paper, so it should be evaluated on Atari or Mujoco or even more complex scenarios such as MineCraft or SMAC to be convincing. **Even author noted in Line 57 in the paper that multi-level skill hierarchies are essential for solving complex tasks. Why are there no experiments on such complex scenarios? In my opinion this paper won't be accepted in a top-tier conference while you don't have a non-toy problem evaluation.**
> >
> > 4. Some issues.
> > - If options are used to solve complex problems, then they should be diverse. In this case, the proposed approach that options control agents move between adjacent state spaces does not make sense, because such options cannot adapt to scenarios that require options to extensively explore different action spaces.
> > - The proproposed approach is built on the assumption that the Louvain method successfully generates good clustering results. If the Louvain method fails or generates clustering results with low quality, the proproposed approach will fail. Such situations are prevalent in the experimental scenarios commonly used in current works. The state-action features of these scenarios do not have characteristics  conducive to unsupervised clustering, and cannot generate appropriate clustering results by simply applying the Louvain method.
> > - If the collected transition graph is only a small part of that in the entire senario, then the results do not provide meaningful guidance for solving the entire problem.

---

> > > ### Comment · Reviewer_UjdQ · 2023-08-18
> > >
> > > - The proposed approach combing lower actions into higher level may not be efficient. It may be less efficient than directly learning actions.
> > >
> > > [1] Şimşek, Özgür, and Andrew Barto. "Skill characterization based on betweenness." Advances in neural information processing systems 21 (2008).
> > >
> > > [2] Konidaris, George, and Andrew Barto. "Skill discovery in continuous reinforcement learning domains using skill chaining." Advances in neural information processing systems 22 (2009).
> > >
> > > [3] Vigorito, Christopher M., and Andrew G. Barto. "Intrinsically motivated hierarchical skill learning in structured environments." IEEE Transactions on Autonomous Mental Development 2.2 (2010): 132-143.
> > >
> > > [4] Hengst, Bernhard. "Discovering hierarchy in reinforcement learning with HEXQ." ICML. Vol. 19. 2002.
> > >
> > > [5] Thrun, Sebastian, and Anton Schwartz. "Finding structure in reinforcement learning." Advances in neural information processing systems 7 (1994).
> > >
> > > [6] Mugan, Jonathan, and Benjamin Kuipers. "Autonomously learning an action hierarchy using a learned qualitative state representation." (2009).
> > >
> > > [7] Konidaris, George Dimitri, and Andrew G. Barto. "Efficient Skill Learning using Abstraction Selection." IJCAI. Vol. 9. 2009.
> > >
> > > [8] Firby, R. James, and Marc G. Slack. "Task execution: Interfacing to reactive skill networks." AAAI Spring Symposium. 1995.
> > >
> > > [9] Bagaria, Akhil, Jason K. Senthil, and George Konidaris. "Skill discovery for exploration and planning using deep skill graphs." International Conference on Machine Learning. PMLR, 2021.
> > >
> > > [10] do Couto, Jorge Manuel Ferreira. "Learning from Demonstration using Hierarchical Reinforcement Learning for Robotic Skill transferability." (2022).
> > >
> > > [11] Blondel, Vincent D., et al. "Fast unfolding of communities in large networks." Journal of statistical mechanics: theory and experiment 2008.10 (2008): P10008.
> > >
> > > [12] De Meo, Pasquale, et al. "Generalized louvain method for community detection in large networks." 2011 11th international conference on intelligent systems design and applications. IEEE, 2011.
> > >
> > > [13] Traag, Vincent A. "Faster unfolding of communities: Speeding up the Louvain algorithm." Physical Review E 92.3 (2015): 032801.
> > >
> > > [14] Dugué, Nicolas, and Anthony Perez. Directed Louvain: maximizing modularity in directed networks. Diss. Université d'Orléans, 2015.
> > >
> > > [15] Bhowmick, Ayan Kumar, et al. "Louvainne: Hierarchical louvain method for high quality and scalable network embedding." Proceedings of the 13th international conference on web search and data mining. 2020.
> > >
> > > [16] Moradi, Parham, Mohammad Ebrahim Shiri, and Negin Entezari. "Automatic skill acquisition in reinforcement learning agents using connection bridge centrality." Communication and Networking: International Conference, FGCN 2010, Held as Part of the Future Generation Information Technology Conference, FGIT 2010, Jeju Island, Korea, December 13-15, 2010. Proceedings, Part II. Springer Berlin Heidelberg, 2010.
> > >
> > > [17] Rad, Ali Ajdari, Martin Hasler, and Parham Moradi. "Automatic skill acquisition in Reinforcement Learning using connection graph stability centrality." Proceedings of 2010 IEEE International Symposium on Circuits and Systems. IEEE, 2010.
> > >
> > > [18] Şimşek, Özgür, Alicia P. Wolfe, and Andrew G. Barto. "Identifying useful subgoals in reinforcement learning by local graph partitioning." Proceedings of the 22nd international conference on Machine learning. 2005.
> > >
> > > [19] Kazemitabar, Seyed Jalal, and Hamid Beigy. "Using strongly connected components as a basis for autonomous skill acquisition in reinforcement learning." Advances in Neural Networks–ISNN 2009: 6th International Symposium on Neural Networks, ISNN 2009 Wuhan, China, May 26-29, 2009 Proceedings, Part I 6. Springer Berlin Heidelberg, 2009.

---

> > > ### Author Response · Authors · 2023-08-20
> > > **Response to Points 1 and 2**
> > >
> > > We thank the reviewer for their continuing engagement with the paper. Below we provide a point-by-point response to the concerns raised. We believe that we have addressed all of the main concerns.
> > >
> > > (1) The innovation in the paper is the use of an existing method (Louvain Algorithm) in a new context (temporal abstraction in reinforcement learning). **This is an established and valuable form of contribution to science.** Our use of the Louvain Algorithm brings something new and useful to temporal abstraction in reinforcement learning, achieving something that is not yet possible with ANY existing approach: autonomous specification of a multi-level action hierarchy with no human input.
> > >
> > > Organising the state-action space of an agent into a skill hierarchy is no easy task. Louvain skills do it elegantly and successfully. This is an important result for the field.
> > >
> > > Please note the following:
> > >
> > > (a) Yes, some existing characterisations of skill hierarchies use the state-transition graph, and some of these methods use graph partitioning; we note this clearly in the paper. There is a fundamental limitation shared by all existing graph-based methods: they lead to skill hierarchies with only **a single level** above primitive actions. The necessity of **multi-level** abstraction is clear; it is noted frequently in the literature (e.g.; Barto, Singh, Chentanez, ICDL 2004; Singh, Barto, & Chentanez, NeurIPS 2004).
> > >
> > > (b) The reviewer wrote: *“However, the Louvain algorithm has long been proposed with many improvement works[11-15]”*. This is a point that the reviewer raised earlier; we have already responded to it: These later works are not relevant in the context of temporal abstraction in reinforcement learning. If the reviewer knows differently, we would welcome the new information, but the reviewer needs to explain which particular later development is useful in our context and why. It is also worth noting that any existing/future improvements to the Louvain algorithm, where relevant and useful, can be **directly** incorporated into our approach as long as the output of the algorithm is a cluster hierarchy.
> > >
> > > In short, we argue that this objection is not valid. We note that it is not possible to further respond to it without further details from the reviewer.
> > >
> > > (c) The method used to train option policies is detailed in the Supplementary Material (Appendix F), and also in our discussion with Reviewer 4HvS. This is just one example of how Louvain option policies can be trained; other approaches exist. Our analysis focused on how useful the Louvain options are when they are available to an agent; therefore, precisely how the option policies were trained did not matter as long as the learned policies were accurate.
> > >
> > > (d) Factual correction: Cited works [16] and [17] are not based on graph partitioning, as claimed by the reviewer. They use graph centrality measures to find bottleneck states and produce skills that efficiently take the agent to these states.
> > >
> > > ---
> > >
> > > (2) We do not claim to contribute a novel hierarchical graph partitioning algorithm. On the contrary, we clearly state throughout the paper that we use the Louvain algorithm (Blondel et al., 2008) to perform hierarchical graph partitioning. Please see our response to point 1 above for the innovation in the paper.

---

> > > ### Author Response · Authors · 2023-08-20
> > > **Response to Point 3**
> > >
> > > (3) We argue that the entire paper supports our claim, including the following results:
> > >
> > > (i) In a wide variety of domains, Louvain skills form a multi-level hierarchy that matches human intuition well (Figure 2).
> > >
> > > (ii) Agents using Louvain skills consistently learn faster than agents using only primitive actions and agents using skills produced by existing approaches (Figure 3).
> > >
> > > (iii) Agents learn more quickly when Louvain skills are arranged into a multi-level hierarchy compared to when each skill calls primitive actions directly (Figure 4).
> > >
> > > (iv) Agents learn more quickly with the full Louvain hierarchy compared to when each level of the skill hierarchy is used in isolation (Figure 4).
> > >
> > > (v) The proposed approach continues to produce good skill hierarchies in environments with over 1 million states (Figure 5b). The Louvain agent continues to have a clear and substantial advantage over other approaches in the largest domain we tested (Figure 5c).
> > >
> > > (vi) We explore two possible paths towards discovering Louvain skills incrementally, while an agent is interacting with its environment, with positive results (Figure 6).
> > >
> > > (vii) We illustrate how a Louvain cluster hierarchy can be produced in a problem with a continuous state space (Figure 7).
> > >
> > > ***On theoretical results:*** (1) Louvain skills have a solid basis in graph theory. (2) We have already noted in our earlier response that any existing convergence results in reinforcement learning continue to hold because learning continues in primitive-action space. This is well known so we did not see the need to include it in the paper. We can do so if the reviewer believes it to be useful. (3) The reviewer is asking for theoretical results but without specifying what they should/could be. Can the reviewer be more specific? Precisely what type of theoretical analysis would be applicable, feasible, and useful?
> > >
> > > ***Scenarios used in the experiments:*** First, we note that it is entirely irrelevant how old the scenarios are; what matters is how useful they are in answering scientific questions posed in the paper. Therefore, “age” is not a valid scientific objection to the scenarios used. Secondly, the reviewer’s comment misses the following: **In these scenarios, our approach achieves something that is not achieved by ANY existing approach: autonomous specification of a multi-level action hierarchy with no human input.** Furthermore, empirical results are presented on a diverse set of scenarios, and they consistently show positive results for the proposed approach. Thirdly, something that is a key difficulty for the field as a whole — scaling up graph-based skill discovery algorithms to very large domains — cannot be the basis of rejecting a new paper that brings forward a new idea whose utility, as well as comparative strengths to existing methods, is clearly demonstrated in smaller domains. We expand on this third point below.
> > >
> > > Constructing graphical representations for very large environments – such as those with continuous (e.g., MuJoCo) or high-dimensional (e.g., Atari and SMAC) state spaces – is a key open problem. Addressing this problem is not our focus. We are asking a more fundamental question: What is a useful skill hierarchy? Scaling up is a different question; and, importantly, it is an **orthogonal** question: Any promising future graph construction method for such domains can be **immediately** incorporated into the proposed approach to produce Louvain skills. It is also noteworthy that any solution to the scaling-up question will benefit not only our approach but basically all other graph-based approaches to skill discovery (as well as other areas of reinforcement learning). Scientific progress is achieved by divide-and-conquer; not all questions can or should be addressed in one conference paper.

---

> > > ### Author Response · Authors · 2023-08-20
> > > **Response to Points 4.1, 4.2, and 4.3**
> > >
> > > *(4) Some issues.*
> > >
> > > *(4.1) If options are used to solve complex problems, then they should be diverse. In this case, the proposed approach that options control agents move between adjacent state spaces does not make sense, because such options cannot adapt to scenarios that require options to extensively explore different action spaces.*
> > >
> > > It is not clear what the reviewer means by *“diverse”*. As noted in our earlier response, Louvain skills are diverse in their initiation sets, policies, termination conditions, and temporal reach. They provide good coverage of the entire state space. It is unclear what other source of diversity one could want from a given class of skills.
> > >
> > > We also do not understand what the reviewer means by *“scenarios that require options to extensively explore different action spaces”*. In particular, what does the reviewer mean by *“different action spaces”*? Can they please express it in the mathematical notation used in the paper? e.g., $A(s)$ is the set of actions available from state $s$ in a given MDP. What exactly are the *“different action spaces”* the reviewer refers to?
> > >
> > > In short, it is not possible to respond to a concern that is not clearly stated. Can the reviewer please further explain their concern? It would help us understand the concern if they are able to point to any existing approach to skill discovery that addresses this concern.
> > >
> > > ---
> > >
> > > *(4.2) [...] If the Louvain method fails or generates clustering results with low quality, the proposed approach will fail. Such situations are prevalent in the experimental scenarios commonly used in current works. The state-action features of these scenarios do not have characteristics conducive to unsupervised clustering, and cannot generate appropriate clustering results by simply applying the Louvain method.*
> > >
> > > (i) The reviewer wrote *“Such situations are prevalent in the experimental scenarios commonly used in current works.”* How prevalent? What are these works? Can the reviewer please provide references so that we can look at the evidence? On the contrary, existing work suggests that, if there exists modular structure in a given network, the Louvain algorithm usually identifies it (e.g., see Lancichinetti & Fortunato, 2009). This is also what we found in our experiments, as reported in the paper.
> > >
> > > (ii) It is possible, even likely, that no single skill characterisation will work perfectly in all possible scenarios. This is not a problem. An agent is not limited to using a single approach to skill discovery; it can (and probably should) use multiple different approaches. It is useful to know what type of environments each discovery algorithm is well suited to. Some problems will have strong modular structure, and others may not; Louvain skills will be better-suited to the former cases than the latter.
> > >
> > > (iii) Even in the absence of modular structure, we still observed that the proposed approach produced useful and intuitively-appealing skills. For instance, consider the Level 2 skills produced in Rooms (Figure 2, top row, third column). The skills for moving between the clusters within each room clearly enable efficient low-level navigation of the state space, despite the fact that the internal structure of each room is uniform.
> > >
> > > ---
> > >
> > > *(4.3) If the collected transition graph is only a small part of that in the entire scenario, then the results do not provide meaningful guidance for solving the entire problem.*
> > >
> > > (i) As soon as any state is visited the very first time, it can be added to the state-transition graph. If unknown parts of the state-transition graph are relevant for solving the problem, the agent will visit those states sooner or later.
> > >
> > > (ii) Future work can explore how to generalise graph structure from known parts of the state space to unknown parts of the state space. This is an orthogonal research question that can be explored in isolation from the current paper and would benefit many areas of reinforcement learning beyond skill discovery.

---

> > > ### Author Response · Authors · 2023-08-20
> > > **Response to Point 4.4**
> > >
> > > *(4.4) The proposed approach combing lower actions into higher level may not be efficient. It may be less efficient than directly learning actions.*
> > >
> > > We find this claim surprising and counter-intuitive. We would welcome any reasoning and evidence that the reviewer can provide to support their claim.
> > >
> > > We argue the opposite, with clear reasoning and evidence: Building multi-level skill hierarchies leads generally to increased learning efficiency. When an agent executes a skill, it learns (through reinforcement learning algorithms) about the consequences of executing not only that skill but also ALL lower-level skills (and, ultimately, primitive actions) that the skill calls upon when executing. So multi-level skills introduce a clear learning efficiency in this way. This is supported by the results in our paper: agents with Louvain skills arranged as a multi-level hierarchy – where lower-level skills are composed into higher-level skills – consistently learned more efficiently than agents using “flat” Louvain skills which call primitive actions directly (Figure 4).
> > >
> > > Furthermore, when learning option policies, it is more economical to form new higher-level skills by composing already-trained lower-level skills than it is to start training from scratch using only primitive actions. As a simple example, if an agent already knows how to leave the room, it can use this knowledge (i.e., skill) when learning how to leave the building.
> > >
> > > ---
> > >
> > > Once again, we thank the reviewer for their continued attention to the paper and hope that our comments are useful in evaluating the paper.

---

### Official Review · Reviewer_7zy2 · 2023-07-06

**Soundness:** 3 good
**Presentation:** 3 good
**Contribution:** 2 fair
**Rating:** 5
**Confidence:** 4

**Summary:**

The paper proposes a graph-partitioning-based method to automatically learn multi-level hierarchies of skills at varying timescales in reinforcement learning. The method assumes existence of a complete state transition graph and applies Louvian algorithm to create a hierarchy of state clusters. By merging states, the algorithm creates state partitions that maximize modularity, which measures the quality of partitions based on strong connections within clusters and weak connections between them.

The work considers an important unsolved problem of automatically creating multi-level skill hierarchies. The paper is written well, however, it needs more clarity. Some details regarding the assumptions made and the scope of problems need to be clarified right from the beginning to improve quality. The empirical evaluation could be improved by diversifying the choice of the domains because four out of six domains are extremely similar navigation domains, and clearly justifying the choice of baselines and their input requirements.

**Strengths:**

The number of levels in a hierarchy do not need to be predefined, unlike existing methods. The proposed method automatically finds the levels in the hierarchy using no gain in modularity as a stopping condition.

**Weaknesses:**

(I) The proposed method relies on concrete state transition graph and then uses hierarchal clustering based algorithm to iteratively merge states and produce a hierarchy of clusters. Hence, the method might not scale well to problems with large state space as the memory requirement would increase exponentially and learning a complete state transition graph would require exploration of the complete state space/high number of samples.

(II) The assumption regarding the availability of a complete state space graph and the scope of the problems (discrete state space, low-dimensional, etc.) the method can handle need to be stated clearly.

(III) The empirical evaluation could be improved by diversifying the selection of domains. It is unclear how the approach would perform in more complex decision-making tasks.

(IV) An analysis of comparison of the number of samples required by each baseline to learn a hierarchy and option policies is needed.


Minor:

(I) line 20: has -> have

(II) line 112: the partition -> the partition c_i is



**Questions:**

(I) Can you explain "we use each of the h partitions to define a single layer of skills, resulting in a hierarchy of h levels.."? It is not clear how total number of partitions is directly affecting the number of the layers.

(II) How can state abstraction be employed using the proposed method?

(III) Can the current method handle continuous state space problems?

(IV) How does the proposed work relate to [1]?

(V) In empirical evaluation, how is the office world (without any objects) different from the maze world domain?

(VI) Can you comment about scalability of the approach to handle domains with large state spaces?

(VII) Do the baselines assume a state transition graph or start learning from scratch? Does the learning performance in Fig 3 include the samples required to learn the hierarchies? If not, how many samples were required to learn the hierarchies by different methods along with the samples required to learn the option policies?


References:

[1] Fox, R., Krishnan, S., Stoica, I. and Goldberg, K., 2017. Multi-level discovery of deep options. arXiv preprint arXiv:1703.08294.


**Limitations:**

(Included in summary, weaknesses)

---

> ### Author Rebuttal · Authors · 2023-08-09
>
> Thank you for the time and attention you have given to our paper.
>
> **Q1: Can you explain “we use each of the h partitions to define a single layer of skills, resulting in a hierarchy of h levels”?**
>
> We will give an example using Figure 2 in the paper. In Rooms, the algorithm produces 4 partitions of the state space, as shown in the top row of Figure 2. Consider the "Level 4" partition. This partition defines level 4 of the skill hierarchy, which has 6 skills: 1) from purple to green, 2) from purple to yellow, 3) from green to yellow, and so on. Similarly, the "Level 3" partition defines level 3 of the skill hierarchy, which has 8 skills: 1) from yellow to blue, 2) from yellow to green, 3) from blue to red, and so on. Similarly, the Level 2 and Level 1 partitions, respectively, define level 2 and level 1 of the skill hierarchy. (At the very bottom of the skill hierarchy, which could be called level 0, we have the primitive actions.)
>
> **Q2: How can state abstraction be employed using the proposed method?**
>
> The most natural place to introduce state abstraction would be when constructing the state-transition graph. Instead of a concrete state, each node could represent an *abstract* state based on some learned representation of the environment, and the Louvain algorithm could be applied to this abstract state-transition graph.
>
> We also note that how to employ state abstraction is an important open problem for graph-based skill discovery generally. Any promising methodology could be directly incorporated into the discovery of Louvain skills.
>
> **Q3: Can the current method handle continuous state space problems?**
>
> Preliminary results in the paper suggest that it can (see Figure 7). We explored one possible approach for building a state-transition graph to represent a continuous domain, and we partitioned the graph by using the Louvain algorithm to form the basis of a concrete set of Louvain skills.
>
> **Q4: How does the proposed work relate to Fox et al. (2017)?**
>
> Fox et al. (2017) propose DDO, an imitation learning method that extracts option hierarchies from demonstration trajectories. It uses policy-gradient methods to train option policies and termination conditions that are most likely to generate given demonstration trajectories. This is fundamentally very different to our approach.
>
> DDO requires demonstration trajectories; our approach does not. With DDO, both the number of hierarchy levels and the number of options per level need to be specified by the system designer ahead of training. In contrast, our proposed approach automatically finds a suitable number of hierarchy levels and automatically produces a suitable number of skills at each level. Furthermore, using an incremental version of our method, the number of hierarchy levels and skills per level can evolve during training.
>
> **Q5: How is the office world (without any objects) different from the maze world domain?**
>
> The dynamics of Office are the same as the other gridworlds (except for the elevator tile). We designed Office to be able to explore the scaling properties of our approach as the number of states increases (see figures 5b and 5c).
>
> **Q6: Can you comment about scalability of the approach (…)?**
>
> Our main contribution is a novel characterisation of a useful skill hierarchy based on the concept of modularity. This contribution, we believe, scales to large state spaces conceptually.
>
> As the size of the state space increases, we will need to consider different ways of representing the graphical structure of the environment (direct use of the state-transition graph will not be useful). We expect that future work will identify and explore different approaches with different strengths and weaknesses under different conditions.
>
> The Louvain algorithm itself scales well to large graphs. Blondel et al. (2008) successfully applied it to graphs with millions of nodes and billions of edges and observed its time complexity to be linear in the number of graph edges. We also successfully applied it to versions of Office with over 1 million states.
>
> **Q7: Do the baselines assume a state transition graph or start learning from scratch? How many samples were required to learn the hierarchies by different methods along with the samples required to learn the option policies?**
>
> All baselines assumed access to the complete state-transition graph.
>
> Our analysis focused on the comparative benefits of the learned skills – because the fundamental question we are asking is *what is a good skill hierarchy?* – so we did not measure the number of samples required by the different methods. All that mattered was that the skills were accurately learned in each case.
>
> Please note that there is no single number of samples required by each algorithm: more samples generally lead to better learning of the underlying concept; so, for each algorithm, there is a range (of number of samples) for which the learned skills are useful to varying degrees.
>
> **The assumption regarding the availability of a complete state space graph and the scope of the problems (discrete state space, low-dimensional, etc.) the method can handle need to be stated clearly.**
>
> Thank you, we will make sure that the assumptions in the different parts of the paper are clearly stated.
>
> When testing the utility of the general concept (Louvain skills), we assume that the graph is available. When exploring incremental learning algorithms, we do not make that assumption.
>
> The most direct use of the Louvain algorithm assumes discrete state and action spaces. But our characterisation of a useful skill hierarchy (based on modularity) is a general concept that can be used with abstract or approximate versions of a state-transition graph. Fundamentally, all that is required is a graphical representation of an agent's interaction with its environment – given or learned, exact or approximate, complete or partial, etc. – that encodes what is known about the connective structure of this interaction.

---

### Official Review · Reviewer_4HvS · 2023-07-07

**Soundness:** 3 good
**Presentation:** 3 good
**Contribution:** 3 good
**Rating:** 7
**Confidence:** 4

**Summary:**

Given a graph of connectivity of the state space of an environment, this paper proposes to use a hierarchical clustering technique, the Louvain algorithm, to provide reinforcement learning with a multi-level skill representation.  This allows policies to take actions at variable time scales, and experimental results show that the identified clusters allow faster learning in the domains studied.

**Strengths:**

- Clarity of presentation.  The graph clustering algorithm is motivated and described very clearly.  The experiments are also well described.  The visualizations of the hierarchical results are informative and easy to interpret.

- Comparison with baselines.  The paper describes five alternative graph-based approaches, and shows superior performance of the Louvain-based method.

- Initial explorations into some limitations.  The paper points out that the method requires full knowledge of the state space and it's graph representation.  Initial results explore what happens when the clustering is learned online with interaction in the environment.  The paper also explores extending the method to continuous state spaces.

- Thorough discussion section.

**Weaknesses:**

- Although the description of the Louvain algorithm and most of the rest of the paper is quite thorough and well presented, the method for training a policy using the hierarchical clustering output is left comparatively sparse.  Especially since the cited method may not be readily familiar to a modern audience.

**Questions:**

The paper is very clear.  Other than the weakness mentioned above, I do not feel I have pressing questions.

**Limitations:**

Limitations are more than adequately addressed, as mentioned in the strengths section above.

---

> ### Author Rebuttal · Authors · 2023-08-09
>
> Thank you for the time and attention you have given to our paper.
>
> **Although the description of the Louvain algorithm and most of the rest of the paper is quite thorough and well presented, the method for training a policy using the hierarchical clustering output is left comparatively sparse. Especially since the cited method may not be readily familiar to a modern audience.**
>
> Thank you for pointing out the need for additional clarification. We provided full experimental details in the supplementary material (Appendix F) and will expand on this specific point in the main paper.
>
> Consider a Louvain option for moving from cluster $c_i$ to a neighbouring cluster $c_j$. To train the policy of this option, macro-Q learning was applied to the following task: the agent starts in a random state in cluster $c_i$; it receives a reward of $-0.01$ at each decision stage until it reaches a state in cluster $c_j$, where it receives a reward of $+1.0$ and the episode terminates. After many episodes of training, this produces an option policy that efficiently takes the agent from any state in $c_i$ to the cluster $c_j$. The policies of options at level $i$ of the hierarchy, $i > 1$, call only the options from level $i-1$; the policies of options at level 1 call only the primitive actions.
>
> This is just one example of how Louvain option policies can be trained. Other approaches exist. Our analysis focused on exploring how useful Louvain options are when available to an agent; therefore, precisely how the option policies were trained did not matter as long as the learned policies were correct.

---

> > ### Comment · Reviewer_4HvS · 2023-08-16
> >
> > Thank you for your response, I agree that the exact method of training the options is not a main point of the paper.  Thanks for pointing out where the set-up was described nonetheless.

---

### Official Review · Reviewer_cG6b · 2023-07-07

**Soundness:** 3 good
**Presentation:** 4 excellent
**Contribution:** 3 good
**Rating:** 7
**Confidence:** 4

**Summary:**

This paper addresses the problem of discovering a hierarchy of skills based on the state-transition graph. The proposed approach builds on an existing method called the Louvain algorithm, which creates a hierarchy of state clusters; i.e., the lowest level will place nearby states in the same (small) cluster, the second level will place those smaller clusters in bigger clusters, and so on. The clusters are defined using a measure of modularity (nodes within a cluster have dense connections, but there are sparse connections between different clusters). The skills are then defined using the options framework, and their behavior is to take the agent from any state within the cluster to the adjacent clusters. The empirical evaluation presented indicates that the method is capable of finding such hierarchies of skills and that, in some cases, using these skills yields a better performance when learning to solve a task, when compared with existing methods. Additionally, the empirical evaluation demonstrates the method's efficacy under scenarios with continuous state spaces (assuming a discretization method) and under scenarios with many states. Finally, the authors present preliminary results for a possible extension of the method to an interactive case where the transition graph is also learned dynamically instead of being assumed available originally.

**Strengths:**

- The paper addresses a very interesting, novel, and important problem: to autonomously identify a hierarchy of skills with multiple levels (not only two as in most previous work), without specifically setting the number of levels it should have.
- The proposed method is intuitive and has a different/creative look at a problem that has not been solved yet.
The paper is very clearly written, with a very good display of the empirical results, analyzing both qualitative and quantitative aspects of the learned skills.
- The experiments conducted indicate that the hierarchies learned are presenting the desired behavior (i.e., clusters well connected within themselves but not well connected between each other).
- The domains used in the empirical evaluation cover a wide range of scenarios: smaller, simpler MDPs, but also MDPs with many states and even an MDP with discretized continuous states. This helps demonstrate the effectiveness of the method at different levels of complexity.


**Weaknesses:**

- The method requires complete knowledge of the state-transition graph, which is often not available in real-life applications. However, I believe the paper still presents an important contribution and a first step towards solving the problem of finding multi-level hierarchies of skills. Additionally, the authors present preliminary results for an incremental version of the algorithm, where the skills are adjusted as the estimated model is learned.
- Without limiting the number of clusters/levels in some way, the algorithm could end up returning a lot of skills, which could make the learning process later on more difficult instead of facilitating it (given the increased action space). The authors mention this in the paper and propose one possible improvement (to ignore lower-level clusters that contain a small number of states).
- The method finds skills using only the transition graph, not the reward function. Thus, it would not be as useful in tasks with a large state space but where only a small subset of it is used for solving the task.

**Questions:**

- Why did the authors select Q-learning as the baseline for primitive actions instead of a more recent, and better-performing, algorithm?
- It seems like the first two plots of Figure 4 do not present the results for Level 4. Could the authors please clarify this?
- Currently, as per my understanding, all edges in the graph have the same weight. This would imply that when finding the skills in non-deterministic MDPs, a transition that happens with a 10% probability would have the same importance as one that always happens. Do the authors have an insight into how the algorithm could be changed to consider the transition probabilities?

**Limitations:**

The authors discuss the limitations of the work. No major concerns.

---

> ### Author Rebuttal · Authors · 2023-08-09
>
> Thank you for the time and attention you have given to our paper.
>
> **Q1: Why did the authors select Q-learning as the baseline for primitive actions instead of a more recent, and better-performing, algorithm?**
>
> The main point of the baseline algorithm is to allow a comparison of the agent's performance with and without Louvain skills. Louvain skills themselves are agnostic to any particular hierarchical reinforcement learning algorithm, so we could have performed our analysis with any algorithm as long as it had a primitive counterpart we could use for our comparison.
>
> Louvain skills are based on the graphical structure of an agent's interaction with its environment; so the most natural test of their fundamental utility is in environments with discrete states and discrete actions. We used Q-learning as the baseline because (1) it is a suitable algorithm for such environments, and (2) it is the direct primitive counterpart to macro-Q learning and intra-option learning, which we used to train the hierarchical agents.
>
> **Q2: It seems like the first two plots of Figure 4 do not present the results for Level 4. Could the authors please clarify this?**
>
> We used a single legend for all three plots in Figure 4. We now see that this created confusion and will remedy it. In the domains shown in the first two plots (Rooms and Taxi), the Louvain skill hierarchy contained only three levels, so there was no "Level 4" agent.
>
> **Q3: Currently, all edges in the graph have the same weight. This would imply that when finding the skills in non-deterministic MDPs, a transition that happens with a 10% probability would have the same importance as one that always happens. Do the authors have an insight into how the algorithm could be changed to consider the transition probabilities?**
>
> For stochastic MDPs, it would be appropriate to weight edges according to the probability of the underlying transition. Many edge-weighting schemes have been proposed in the graph-based skill discovery literature (e.g., see Section 3.1, Metzen (2013)). One sensible approach is to assign an edge from state $u$ to state $v$ a weight of $\sum_{a \in A(u)}P(u,a,v)$. These probabilities could be based on the true transition probabilities, if known, or could otherwise be estimated from experience.
>
> The definition of modularity naturally handles weighted graphs, so the Louvain algorithm (and, therefore, the proposed method) will be able to use this information. The higher the probability of transitioning between two states, the higher the weight of the edge between them, and the higher the likelihood that the Louvain algorithm will place these two states in the same cluster.
>
> **Without limiting the number of clusters/levels in some way, the algorithm could end up returning a lot of skills, which could make the learning process later on more difficult instead of facilitating it (given the increased action space).**
>
> Making a large set of options available to the agent can indeed harm learning. But Louvain skills have a number of useful properties that limit the impact of this problem.
>
> First, we observe empirically that the depth of the Louvain skill hierarchy grows very slowly with the size of the state space. Figure 4b shows that the hierarchy depth increases from 5 levels in a version of Office with 1000 states to 8 levels in a version of Office with over 1 million states. This aligns with existing results reported in the literature, such as from Blondel et al. (2008), who find that the Louvain algorithm produced 6 levels when applied to a graph of a social network with over 2 million nodes.
>
> Secondly, two parameters influence the depth of the Louvain skill hierarchy: the resolution parameter, $\rho$, and the mean cluster size threshold, $c$.
> Higher values of $\rho$ punish overall cluster size and inter-cluster edges more harshly, leading the Louvain algorithm to run for fewer iterations and produce fewer partitions. The result is a skill hierarchy with fewer levels.
> On the other hand, $c$ impacts the lowest level of the hierarchy — the algorithm discards partitions containing clusters that are (on average) smaller than $c$. Higher values of $c$ lead to more partitions being discarded, resulting in a skill hierarchy with fewer levels.
>
> Thirdly, although many skills may be defined at each level of the hierarchy, a relatively small number of them will be available in any given state. This is because Louvain skills have restricted initiation sets: a Louvain skill navigating from some source cluster to some target cluster is available only in the states of the source cluster. For example, in Rooms, at level 2 of the hierarchy, at most 3 skills are available from each state (See Figure 2, top row, third column).
>
> Finally, arranging Louvain skills into a multi-level hierarchy allows an agent to learn about multiple skills at the same time. When an agent is executing a skill, it learns about the consequences of executing not only that specific skill but also of any of the lower-level skills (and, ultimately, primitive actions) that the skill calls upon while executing.
>
> **The method finds skills using only the transition graph, not the reward function. Thus, it would not be as useful in tasks with a large state space but where only a small subset of it is used for solving the task.**
>
> Incorporating the task reward when producing Louvain skills would be an interesting avenue for future work. But Louvain skills as defined in the paper, based purely on the connectivity of the state-transition graph, are still useful. In large state spaces, in the absence of further information on where the rewards may be, they will allow efficient exploration of the state space. In our experiments, Louvain skills helped learning efficiency the most in the largest domain we tested (Figure 5c).

---

> > ### Comment · Reviewer_cG6b · 2023-08-21
> >
> > Thank you for your thorough response. After reading it, and the other reviews and discussion, I tend to maintain my original score. I agree that there are limitations and that the idea might be seen as simple, but the limitations are discussed in the paper and I believe this is a valuable contribution to the field.

---

### Decision · Program_Chairs · 2023-09-21

**Decision:**

Accept (poster)

**Comment:**

There was near consensus among the reviewers to accept this paper, praising its novelty, intuitive appeal, clarity, and the thoroughness of the discussion of the results. Several reviewers raised a concern regarding the requirement that the state-transition graph is known, to which the authors pointed out that this is only true for a fraction of the experiments, and that the incremental learning algorithms go beyond this restriction. Further, reviewers initially disagreed about the quality of the empirical valuation, with some praising the diversity and breadth of the experimental domains and others calling for further diversity. The discussion largely resolved the primary concerns of most reviewers. Thus, I believe the strengths outweigh the weaknesses, and I recommend this paper be accepted